



# High–Arctic aircraft measurements characterising black carbon vertical variability in spring and summer

Hannes Schulz[1], Heiko Bozem[2], Marco Zanatta[1], W. Richard Leaitch[3], Andreas B. Herber[1],
Julia Burkart[4,a], Megan D. Willis[4], Peter M. Hoor[2], Jonathan P. D. Abbatt[4], and Rüdiger Gerdes[1,5]

[1]AlfredWegener Institute, Helmholtz Center for Polar and Marine Research, Bremerhaven, Germany
[2]Institute of Atmospheric Physics, Johannes Gutenberg-University, Mainz, Germany
[3]Environment and Climate Change Canada, Toronto, Ontario, Canada
[4]Department of Chemistry, University of Toronto, Ontario, Canada
[a]now at: Aerosol Physics and Environmental Physics, University of Vienna, Austria
[5]Physics & Earth Sciences, Jacobs University, Bremen, Germany

**Correspondence:** Hannes Schulz (hannes.schulz@awi.de)

**Abstract.**

The vertical distribution of black carbon (BC) particles in the Arctic atmosphere is one of the key parameters controlling its radiative forcing. Hence, this work investigates the presence and properties of BC over the high Canadian Arctic. Airborne campaigns were performed as part of the NETCARE project and provided insights into the variability of the vertical distributions of BC particles in summer 2014 and spring 2015. The observation periods covered evolutions of cyclonic disturbances to the polar dome that caused and changed transport of air pollution into the High–Arctic, as otherwise the airmass boundary largely impedes entrainment of pollution from lower latitudes. A total of 48 vertical profiles of refractory BC (rBC) mass concentration and particle size, extending from 0.1 to 5.5 km altitude, were obtained with a Single–Particle Soot Photometer (SP2).

Generally, the rBC mass concentration decreased from spring to summer by a factor 10. Such depletion was associated with a decrease of the mean rBC particle diameter, from approximately 200 nm to 130 nm at low altitude. Due to the very low number fraction, rBC particles did not substantially contribute to the total aerosol population in summer.

Profiles analysed with potential temperature as vertical coordinate revealed characteristic variability patterns due to different balances of supply and removal of rBC in specific levels of the stable atmosphere. Kinematic back–trajectories were used to investigate transport pathways into these levels. The lower polar dome was influenced by low–level transport from sources within the cold central and marginal Arctic. During the spring campaign, a cold air outbreak over eastern Europe additionally caused northward transport of air from a corridor over western Russia to Central Asia that was affected by emissions from gas flaring, industrial activity and wildfires. This caused rBC concentrations between about 500 to 1800 m altitude to gradually increase from 32 to 49 ng m$^{-3}$. The temporal development of transport to the level above, at around 2500 m, caused the initially low concentration to increase from <15 ng m$^{-3}$ to 150 ng m$^{-3}$. Despite the higher concentrations in the upper level, significantly less rBC reached the High–Arctic relative to co–emitted CO. A shift in rBC mass–mean diameter, from above 200 nm in the low–level transport dominated lower polar dome to <190 nm at higher levels, indicates that rBC got affected





by wet removal when lifting processes were involved during transport. The summer polar dome had limited exchange with the mid–latitudes. Air pollution was supplied from sources within the marginal Arctic as well as by long–range transport, but in both cases rBC was largely depleted in absolute and relative concentrations. Near the surface, rBC concentrations were $<2\,\mathrm{ng\,m^{-3}}$, while concentrations increased to $<10\,\mathrm{ng\,m^{-3}}$ towards the upper boundary of the polar dome. The mass–mean particle diameter of 132 nm was smaller than in spring. The shape of the summer mean mass–size distribution, however, resembled the spring distribution from higher levels, which was depleted of particles >300 nm due to nucleation scavenging.

Our work provides vertical, spatial and seasonal information of rBC characteristics in the High–Arctic polar dome, offering a more extensive dataset for evaluation of chemical transport models and for radiative forcing assessments than obtained before by any other aircraft campaign.

# 1 Introduction

Climate change in the Arctic is more rapid than on global scale and a significant loss of the summertime sea–ice extent has been observed over the past decades (e.g. Lindsay et al., 2009). The fast progression of change is largely a result of the ice–albedo and temperature feedback (Screen and Simmonds, 2010; Pithan and Mauritsen, 2014). The driving agents of Arctic warming, however, still remain unclear. Recent studies suggest that next to $CO_2$, short–lived climate forcers contribute significantly to the observed warming, but their complex interactions with the Arctic climate system cause high uncertainties (Quinn et al., 2008; Shindell et al., 2012; Yang et al., 2014; AMAP, 2015; Sand et al., 2015). Black carbon (BC) particles, emitted during incomplete combustion of fossil fuels and biomass, are the major light absorbing component of atmospheric aerosol. Bond et al. (2013) concluded that atmospheric BC's interaction with solar radiation induces a global radiative forcing of $+0.71\,\mathrm{W\,m^{-2}}$, which has an uncertainty range of $+0.08$ up to $+1.27\,\mathrm{W\,m^{-2}}$. BC may also affect the distribution, lifetime, and microphysical properties of clouds when particles act as cloud condensation nuclei (e.g. Chen et al., 2010) or, in BC loaded atmospheric layers, cloud properties can change as adjustment to increased temperature and stability (e.g. Lohmann and Feichter, 2001). The aerosol cloud interaction is suspected to significantly impact climate (IPCC 2013), but the overall level of scientific understanding is still low (Bond et al., 2013). The aerosol interactions with solar radiation and clouds not only depend on concentrations, but also on microphysical properties, namely the size distribution and mixing state, of BC particles (Kodros et al., 2018).

Model studies of the Arctic climate system by Flanner (2013) and Samset et al. (2013) emphasise that Arctic surface temperatures have different sensitivities to BC's radiative forcing depending on the altitude where the absorbing aerosol layers are distributed. When absorption and scattering through aerosols occur higher in the atmosphere, it has a dimming effect on the solar radiation reaching the surface. The energy absorbed at higher levels is inefficiently mixed downward and atmospheric stability is even increased, thus BC containing aerosol can cause in a net cooling effect at the surface (MacCracken et al., 1986). On the other hand, thermal radiation from absorbed solar light in the lower parts of the atmosphere can actually contribute efficiently to surface warming. Reflections from the bright, high albedo ice and snow surfaces in the Arctic increase the amount of energy absorbed by aerosol like BC. Aerosol particles are removed from the atmosphere by sedimentation as well as nucleation, impaction and below cloud scavenging (e.g. Kondo et al., 2016). The result within the Arctic is likely a



deposition of BC in ice and snow that can darken the otherwise highly reflective surfaces (Flanner et al., 2008; Tuzet et al., 2017, and references therein). Studies (e.g. Hansen and Nazarenko, 2004; Flanner et al., 2007) suggest that albedo decrease due to deposition can offset the cooling effect through dimming by higher atmospheric aerosol, but the balance of these effects in the Arctic can only be estimated in models as accurately as the distribution of BC is known. However, there is a spread

of more than one order of magnitude amongst different state–of–the–art models as well as between models and observations (AMAP, 2015).

Consequently, in order to provide accurate radiative forcing estimation in the Arctic region, it is necessary to understand what controls the vertical distribution of BC particles in the Arctic atmosphere. Import of polluted air from lower latitudes is controlled by the cold airmass that lies over the Arctic like a dome. The polar dome's vertical temperature structure forces

warmer air from the mid–latitudes to ascend along isentropic surfaces when transported into the polar region, reaching it in layers in the mid and upper troposphere (Stohl, 2006). The polar front between the cold polar and the warmer mid–latitude airmasses, a strong horizontal temperature gradient, acts as transport barrier that is controlling the intrusion of polluted air from southern source regions. Emissions from sources within the cold polar dome are transported through the Arctic at lower altitudes. BC emitted from continental areas in the northern hemisphere is mainly carried poleward by mid–latitude low–

pressure systems and is eventually mixed across the polar front in the systems' warm and cold fronts. These frontal systems, with life–times of 1–2 weeks, are frequently generated and poleward mass transport is continuously induced (Stohl, 2006; Sato et al., 2016). The polar dome retreats northward in the summer and leaves many pollution sources south of the polar front. Increased wet removal (scavenging) of aerosol particles is thought to help maintaining much more pristine conditions throughout the Arctic in summer, compared to winter and spring (Barrie, 1986; Shaw, 1995; Garrett et al., 2011; Tunved

et al., 2013; Raut et al., 2017). This pronounced seasonal variability of BC concentration was observed at ground based High–Arctic measurement sites (e.g. Eleftheriadis et al., 2009; Massling et al., 2015; Sharma et al., 2017), however the near surface air is decoupled from the mid and upper troposphere due to the high stability of the atmosphere (Brock et al., 2011) and these measurements cannot represent variability in the vertical (Stohl, 2006). The concentrations of BC particles in the lower atmosphere might be affected by increasing numbers of local emissions. In fact, as the sea ice retreat makes the Arctic

region more accessible, commercial activities in the marginal Arctic (the sea–ice boundary and boreal forest region), associated with flaring of gas in connection with oil production (Stohl et al., 2013; Evans et al., 2017) and shipping (e.g. Corbett et al., 2010; Aliabadi et al., 2015), are increasing and the possible consequences are an area of current research demand (Law et al., 2017). Models aiming to assess the radiative forcing impact of Arctic aerosol have shortcomings in the representation of concentrations and size distributions as well as their vertical variability, which was partly attributed to incorrect treatment of

scavenging processes in parametrisations (Schwarz et al., 2010b; Liu et al., 2011). Therefore, vertically resolved observations of aerosol mass and size distributions are an important benchmark for chemical transport models.

Nevertheless, measurements of the vertical distribution of BC and its variability in the Arctic atmosphere are very sparse (AMAP, 2015). Aircraft campaigns like ARCPAC, ARCTAS and POLARCAT (Spackman et al., 2010; Brock et al., 2011; Kondo et al., 2011a; Matsui et al., 2011; Wang et al., 2011), PAMARCMIP (Stone et al., 2010), HIPPO (Schwarz et al., 2010b,

2013) and ACCESS (Law et al., 2017; Raut et al., 2017) delivered limited numbers of BC vertical profiles from within the





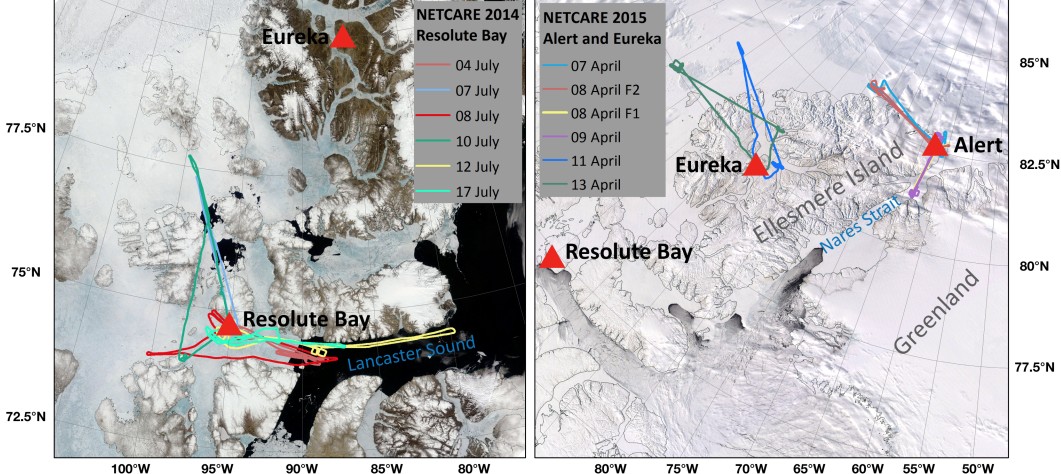

**Figure 1.** Maps of all flight tracks evaluated in this study. The summer measurement flights were operated out of Resolute Bay on the northern shore of Lancaster Sound, southern Canadian Arctic Archipelago (left). The spring campaign operated from Alert and Eureka on Ellesmere Island (right). MODIS satellite true colour images were obtained from https://worldview.earthdata.nasa.gov.

cold polar airmass. To increase the validity and reduce biases of vertical profile measurements, which are the basis to improved system understanding, high–latitude observations at better spatial and temporal resolution are required. Such observations may resolve the internal variability due to weather changes as well as regional characteristics due to the prevailing atmospheric transport pathways with respect to differences between the seasons.

This paper will discuss a set of measurements from the spring and summer aircraft campaigns in the NETCARE project (Network on Climate and Aerosols: Addressing Key Uncertainties in Remote Canadian Environments, http://www.netcare-project.ca). Motivated by the high sensitivity of the mechanisms of BC's radiative forcing in the Arctic climate system on its vertical distribution, the main goal of this study is to characterise the vertical variability of BC concentrations and particle properties in the polar dome, contrasting spring and summer. The campaigns yielded a unique and detailed dataset from within

the polar dome at high latitudes, and covered time scales that give insight in the variability of aerosol distributions due to changes in the meteorological conditions and transport pathways for air pollution in spring and summer.

## 2   Methods and material

### 2.1   Spatial and temporal coverage of research flights

Aerosol observations were carried out with the Alfred Wegener Institute's (AWI) research aircraft Polar 6, a DC–3 fuselage

converted to a turboprop Basler BT–67 (see Herber et al., 2008). This aircraft was specifically modified for polar research and allows flights at relatively low speeds and within an altitude range of 60–8000 m above mean sea level. A constant survey speed of approximately 120 knots and ascent/descent rates of 150 m/min were maintained for vertical profiles.



**Table 1.** Overview of all measurement flights of the NETCARE summer campaign 2014 and and the spring campaign 2015 that are evaluated in this study.

| Campaign | Date | Station | Number of profiles | Campaign | Date | Station | Number of profiles |
|---|---|---|---|---|---|---|---|
| NETCARE summer | 04 07 2014 | Resolute Bay | 4 | NETCARE spring | 07 04 2015 | Alert | 3 |
| NETCARE summer | 07 07 2014 | Resolute Bay | 5 | NETCARE spring | 08 04 2015_F1 | Alert | 4 |
| NETCARE summer | 08 07 2014 | Resolute Bay | 5 | NETCARE spring | 08 04 2015_F2 | Alert | 3 |
| NETCARE summer | 10 07 2014 | Resolute Bay | 5 | NETCARE spring | 09 04 2015 | Alert | 4 |
| NETCARE summer | 12 07 2014 | Resolute Bay | 4 | NETCARE spring | 11 04 2015 | Eureka | 4 |
| NETCARE summer | 17 07 2014 | Resolute Bay | 5 | NETCARE spring | 13 04 2015 | Eureka | 2 |
| | | Total | 28 | | | Total | 20 |

The vertical atmospheric profile measurements were performed during the aircraft campaigns of the NETCARE project in summer 2014 and spring 2015. The summer measurements took place from 4 to 21 July 2014 and the aircraft was based in Resolute Bay (74.68°N, 94.87°W), Nunavut, at the northern shores of Lancaster Sound in the Canadian Arctic Archipelago. During spring, measurements on a total of 10 flights took place from 5 to 21 April 2015 as a traverse through the western Arctic

with four stations: Longyearbyen (78.2°N, 15.6°E), Svalbard; Alert (82.5°N, 62.3°W), Nunavut, Canada; Eureka (80.0°N, 85.9°W), Nunavut, Canada; and Inuvik (68.4°N, 133.7°W), Northwest Territories, Canada. The map in Fig. 1 details all flight tracks from both campaigns analysed in this study, which is a subset of research flights that took place north of the polar front and thus within or above the polar dome. This selection is based on the extent of the polar dome defined in Bozem et al. (2018). Table 1 gives a list of the flights selected for this study.

## 2.2 Measurements

### 2.2.1 Single particle aerosol measurements

A Single Particle Soot Photometer (SP2; 8–channel) by Droplet Measurement Technologies Inc. (DMT, Longmont, CO, USA) was used to detect BC particles. The operation principle and evaluations of the method are given by Stephens et al. (2003); Schwarz et al. (2006); Moteki and Kondo (2010). Briefly, the SP2 is based on the laser–induced–incandescence method: a

concentric-nozzle jet system directs the aerosol sample flow through a high–intensity continuous–wave intra–cavity laser beam at a wavelength of 1064 nm, in which highly absorbing particles, such as BC, are heated to their vaporisation temperature and emit thermal radiation (incandescence). Particles containing a sufficient amount of BC (∼0.5 fg) can absorb enough energy to reach incandescence, which excludes sensitivity to other, less absorptive, material such as organic carbon, brown carbon or inorganic aerosol components. The peak intensity of the emitted thermal radiation, which occurs when the boiling point

temperature of BC is reached, is proportional to the BC mass contained in the particle. Following the terminology defined by





Petzold et al. (2013), the refractory, essentially pure carbon, material detected with the SP2 is hereafter referred to as refractory black carbon (rBC). All other particulates that may be internally mixed with a BC core evaporate at temperatures below the boiling point of BC ($\sim 4000°C$) such that no interference occurs in the quantification of the rBC mass (Moteki and Kondo, 2007).

The incandescence light detector, an avalanche photo–diode with a 350 to 800 nm band–pass filter, used two gain stages. It was calibrated with a Fullerene Soot standard from Alfa Aesar (stock #40971, lot #FS12S011) by selecting a narrow size distribution of particles with a differential mobility analyser upstream of the SP2 (following Schwarz et al., 2010a; Laborde et al., 2012). The mass of these mono–disperse particles was empirically calculated using the relationship between mobility diameter and the effective density of Fullerene Soot (Gysel et al., 2011). The Fullerene Soot calibrations used for the datasets of

the two NETCARE campaigns agreed to within $\pm 10\%$ with each other, ensuring a good degree of comparability between the two campaigns in agreement with the reproducability of SP2 rBC mass measurements evaluated by Laborde et al. (2012). After calibration, the SP2 allowed 100% detection efficiency of particles with mass in the range 0.6 to 328.8 fg, which is equal to 85 to 704 nm mass equivalent diameter ($D_{rBC}$), assuming a void free bulk material density of 1.8 g cm$^{-3}$. The SP2 was prepared for the research flights following the recommendations given in Laborde et al. (2012). Stability of the optical system and laser

power was confirmed during the campaign by measuring mono–disperse polystyrene latex spheres (PSL). An estimated total uncertainty of rBC mass concentrations is 15%, including reproducibility and calibration uncertainty (Laborde et al., 2012) and uncertainties of airborne in situ measurements (e.g. precision of the sample flow measurement). The SP2 was used to obtain rBC mass concentrations ($M_{rBC}$), rBC number–size distribution weighted by particle mass (mass–size distributions MSD) and mass–mean diameters ($MMD$) of rBC particles.

The measured $M_{rBC}$ were not corrected for the mass of particles outside the detection range, and are thus only valid for the range 85 to 704 nm. The contribution of small Aitken mode particles as well as particles larger than 704 nm to the total PM$_1$ rBC mass (mass of particles smaller than 1000 nm) may be significant and the measurements presented here can underestimate the total PM$_1$ mass by variable degrees. Approaches as used by Sharma et al. (2017), to estimate the total PM$_1$ rBC mass by fitting a lognormal distribution to a measured particle MSD, cannot be applied to aircraft measurements since MSD vary with

location and altitude and statistics are insufficient to derive multivariate correction factors. The underestimation of the total PM$_1$ mass due to the contribution of particles smaller than 85 nm were estimated for selected cases to be an additional 4.5% (between 2 and 7%) rBC mass in the summer polar dome, 7.5% (between 4.5 and 8.5%) in the lower spring polar dome and up to 10% (7.8 to 12%) within high concentration pollution plumes. Assuming the SP2 was likely able to count (but not size) all particles between 700 and 1000 nm, an infrequent ($< 30$ particles/flight) underestimation of the PM$_1$ mass due to large

particles occurred in spring in high concentration plumes as well as in the lower atmosphere. No influence of particles larger than 700 nm was apparent for summer conditions.

     The particle number–size distributions and number concentration of the total aerosol (TA) were measured with a DMT Ultra–High Sensitivity Aerosol Spectrometer (UHSAS). As described in Cai et al. (2008), the UHSAS measures the scattered light intensity of individual particles crossing an intra–cavity solid–state laser (Nd$^{3+}$:Y LiF$_4$), operating at a wavelenght of

1054 nm, to evaluate the particle size (under the assumption of the refractive index of PSL particles and spherical shape).



The UHSAS can detect scattering particles over the range 85 to 1000 nm with 95% counting efficiency below concentrations of 3000 cm$^{-3}$ compared to a CPC (Cai et al., 2008). Thus, the instrument covers a size range comparable to the SP2's rBC particle detection range. Fast changes of the aircraft's vertical speed can cause a pressure difference between inlet and exhaust of the instrument and may affect the sample flow measurements, and thus, concentrations reported by standard UHSAS, as

reported by Brock et al. (2011). Instrument modifications were recommended by Kupc et al. (2017). Cross comparisons of data from the non-modified UHSAS used in this study with other aerosol counters (SP2, CPC) have, however, shown no effects. This is likely due to the slow vertical speed maintained during the research flights and the not pressurised cabin of Polar 6 compared to the NOAA WP–3D aircraft used for the measurements reported by Brock et al. (2011). In the following analysis of vertical profile measurements, the number ratio of rBC over TA particles, $R_{\mathrm{numTA}}$, is used to determine layers with enhanced

combustion generated aerosol content.

The air inlet for aerosol sampling was a shrouded inlet diffuser (diameter 0.35 cm at intake point) on a stainless steel tube (outer diameter of 2.5 cm, inner diameter of 2.3 cm) mounted to the top of the cockpit and ahead of the engines to exclude contamination. In–flight air was pushed through the line with a regulated flow rate of approximately 55 L min$^{-1}$, which was estimated to meet nearly isokinetic sampling criteria at survey speed. The transmission efficiency of particles with diameters

between 20 to 1000 nm through the main inlet was approximately unity. The inlet is further discussed by Leaitch et al. (2016). The SP2 and UHSAS shared one bypass line off the main aerosol inlet and sampled with constant 120 ccm (volumetric) and 50 ccm (STP), respectively. A higher flow was maintained in the bypass using a critical orifice and a vacuum pump at its end. The rBC mass and number concentrations presented in this study refer to standard temperature and pressure (STP; 273.15 K, 1013.25 hPa), as the volumetric flow was converted using temperature and pressure readings from the instrument's

measurement chamber.

### 2.2.2   Trace gases

Carbon monoxide (CO) was measured with an Aerolaser ultra fast CO monitor model AL 5002 based on VUV–fluorimetry, using the excitation of CO at a wavelength of 150 nm. The instrument was modified for applying in–situ calibrations during inflight operations. Calibrations were performed on a 15–30 minutes time interval during the measurement flights, using a NIST

traceable calibration gas. The total uncertainty relative to the working standard of 4.7 ppbv (summer) or 2.3 ppbv (spring) can be regarded as an upper limit. See Bozem et al. (2018) for further details of calibrations and corrections. Trace gases were sampled through a separate inlet made of a 0.4 cm (outer diameter) Teflon tubing entering the aircraft at the main inlet and exiting through a rear–facing 0.95 cm exhaust line that provided a lower line pressure. An inlet flow of approximately 12 L min$^{-1}$ was continuously monitored.

Atmospheric BC and CO are often co–emitted from the same combustion sources (Streets et al., 2003), but the relative ratio of the species depends on the combustion type, i.e. fuel types such as biomass or fossil fuel, and combustion conditions, such as flaming, smouldering or (engine) internal. Other than aerosol, which is affected by dry and wet removal mechanisms, CO can be used as a nearly inert combustion tracer within timescales of a few weeks (neglecting possible sources of CO due to biogenic production or by means of oxidation of other trace gases (Gaubert et al., 2016)). The ratio of rBC to CO relative





to their background levels ($R_{CO}$) can be used as an indicator for when rBC particles were depleted by removal processes (Oshima et al., 2012; Stohl et al., 2013). Based on the measured rBC and CO concentrations, the ratio is calculated as $R_{CO} = \Delta rBC/\Delta CO = M_{rBC}/\Delta CO$ (with units $\mathrm{ng\,m^{-3}\,ppbv^{-1}}$). A background completely depleted of rBC is assumed throughout the column as in previous studies (e.g. Moteki et al., 2012; Kondo et al., 2016) due to the short atmospheric lifetime of BC in

the order of days to weeks (Bond et al., 2013). The CO background value is altitude dependent (Fig. A1) and hence defined as the fifth percentile value of all CO mixing ratios observed within defined altitude intervals, following Kondo et al. (2016). $R_{CO}$ is only calculated when $\Delta CO$ exceeded the measurement uncertainty.

### 2.2.3   Meteorological parameters

The meteorological state parameters pressure, humidity and temperature were recorded at 1 Hz resolution with the basic me-
teorological sensor suite and data acquisition of Polar 6. The ambient air temperature was measured with a PT100 type sensor mounted to the aircraft fuselage in a Goodrich/Rosemount 102 EK 1BB housings with deicing facility. Corrections for the deicing heat and adiabatic temperature increase due to pressurisation of the airflow inside the sensor housing (RAM raise and recovery factor) were applied to the temperature readings (following Stickney et al., 1994). The relative humidity (RH) was measured with a Vaisala humidity sensor HMT333 mounted inside a Rosemount housing Model 102 BX, which is also deiced
and similar in its flow characteristics to the housing of the temperature probe. The saturation vapour pressure and RH are corrected with the actual ambient temperature from corrected PT100 readings. The potential temperature was calculated from ambient temperature and the ambient pressure from a static pressure probe.

A Forward Scattering Spectrometer Probe (FSSP), model 100, by Particle Measuring Systems (PMS Inc., Boulder, CO) was used for the measurement of cloud particles. Data from the probe, which was mounted in a canister on a wing pylon, were
analysed in more detail in Leaitch et al. (2016) and contributes to the following analysis as indicator for visible and invisible clouds by an empirically chosen threshold above instrument noise level to the measured cloud particle concentrations (droplets and ice crystals). Aerosol data has been masked when the aircraft was in clouds.

### 2.3   Model weather data and transport pathway analysis

The ERA–Interim re–analysis data (Dee et al., 2011) from the European Centre of Medium–Range Weather Forecasts (ECMWF)
is analysed at certain pressure levels in the form of classical weather maps in order to understand the meteorological situation in the Arctic during the period of our measurement flights. ECMWF operational data is further used to drive the Lagrangian analysis tool (LAGRANTO: Wernli and Davies, 1997; Sprenger and Wernli, 2015) and its kinematic back–trajectories are analysed to estimate the regions of origin for polluted air encountered during the research flights. The model's input data has a horizontal grid spacing of 0.5° with 137 hybrid sigma–pressure levels in the vertical from the surface up to 0.01 hPa. Trajecto-
ries were initialized every 10 seconds from coordinates along the research flight tracks and calculated 10 days back in time. The time series of trajectories along the track of the aircraft were correlated with in–situ measurement values, in particular with rBC and CO concentrations, in order to relate individual features in the vertical profiles to an ensemble of trajectories by the means of threshold filtering. Trajectories fulfilling these criteria are displayed on maps in Sec. 3.4, that also show spatial data





of gas flaring sites from the ECLIPSE (Evaluating the Climate and Air Quality Impacts of Short–Lived Pollutants) emission inventory (Stohl et al., 2015; Klimont et al., 2017) and active fires from the MODIS level 2 satellite product (Giglio et al., 2003) in order to mark potential rBC sources.

# 3 Results

## 3.1 Meteorological overview

With the focus on the polar dome and the vertical distribution of rBC therein, subsets of the flights in spring 2015 and summer 2014 were selected for this analysis, which are based on the variability of the polar dome's position and southern boarder. The structure and extent of the polar dome in both seasons has been evaluated by Bozem et al. (2018), who defined the polar dome based on trace gas gradients measured during the NETCARE campaigns. They found that the polar dome boundary was, on average over the course of the campaigns, located between 61.5°N and 64.5°N in April 2015 and further north, between 68.5°N and 73.5°N, in July 2014. The upper boundary of the dome was found in a potential temperature range between 281.5 and 286.5 K in spring and between 297 and 303 K in summer. An operational estimation of the polar dome's horizontal extent in the mid–troposphere can also be made by locating the maximum latitudinal gradients in geopotential height and temperature, as it indicates the position of the jet stream in the upper troposphere and pressure driven wind systems in the lower troposphere surrounding and stabilising centres of coldest air (Jiao and Flanner, 2016). Maps of these properties (Fig. 2 and 3 for spring and summer, respectively) are shown in this section in order to give an overview of the measurement locations relative to synoptic features of the atmosphere.

The meteorological situation in April 2015 was dominated by a pool of very cold air centred over the Canadian Arctic Archipelago that surrounded the stations Alert and Eureka on Ellesmere Island. The cyclonic flow surrounding the cold air stabilised this system by blocking perturbations of low–pressure systems (Fig. 2). The polar vortex was in a weak state and not well defined. Cold airmasses in the Russian Arctic were cut off from the dome over the Canadian Arctic Archipelago. Near the beginning of the measurement period, a strong low–pressure system caused an outbreak of cold air over Eastern Europe, while warm mid–latitude air moved poleward further west. This synoptic feature affected Alert strongest on 8 April, and its influence was diminishing during the measurements around Eureka 11 to 13 April (Fig. 2). Conditions during all flights were low wind speeds and clear sky with only few, mostly thin clouds (Libois et al., 2016).

The NETCARE summer campaign 2014 operated in an area of the high Canadian Arctic that was situated within the summer polar dome. The first half of the campaign (4 to 12 July) was characterised by a northern influence (Fig. 3). The atmosphere featured a low boundary layer height capped by a distinctive temperature inversion leading to a very stable stratification of the lower troposphere. Prevailing conditions for the research flights were a clear sky, only few or scattered clouds and low wind speeds (Burkart et al., 2017). These conditions gave the opportunity to characterise the summer polar dome in undisturbed conditions, when 6 flights with a total of 28 vertical profiles were conducted in the study area around Resolute Bay on Lancaster Sound (see map in Fig. 1). Starting from 13 July 2014, the weather pattern changed and Resolute Bay got into the transition zone between polar and mid–latitudinal air, as a consequence of a low-pressure system coming from the north–western Beaufort




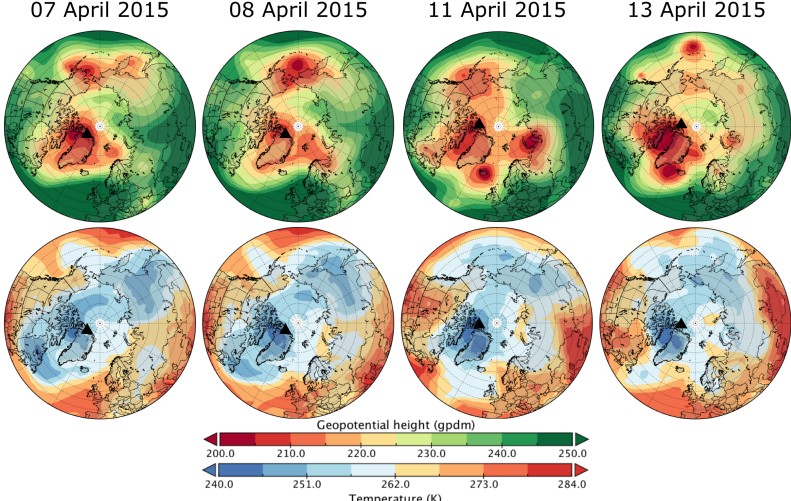

**Figure 2.** Weather charts for the NETCARE spring campaign in April 2015 showing the 750 hPa geopotential height (top) and temperature (bottom) evolution over the duration of the campaign. The stations Alert or Eureka are marked with a triangle for the maps from 7–8 and 11–13, respectively.

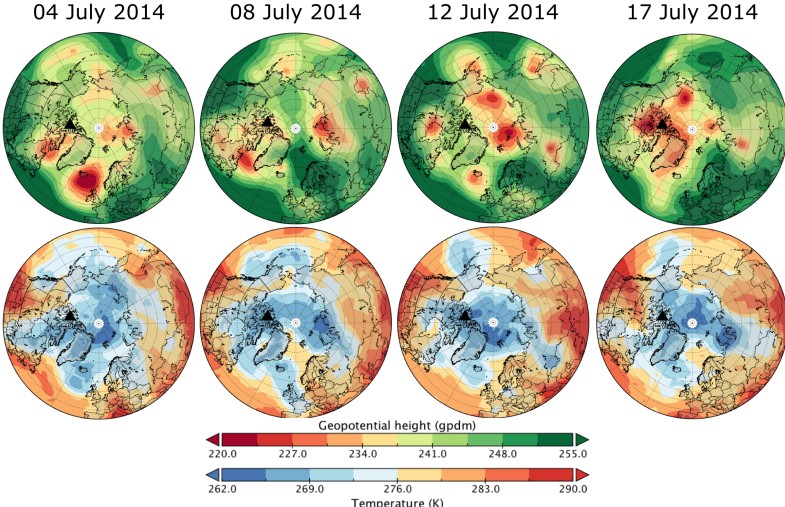

**Figure 3.** Weather charts for the NETCARE summer campaign in July 2014 showing the 750 hPa geopotential height (top) and temperature (bottom) evolution over the duration of the campaign. Resolute Bay is marked with a triangle.

Sea and passing south of Lancaster Sound. Bad visibility due to fog, clouds and precipitation impeded flights until 17 July, on which day the study area shifted back into the cold airmass and a westerly air movement (Fig. 3).





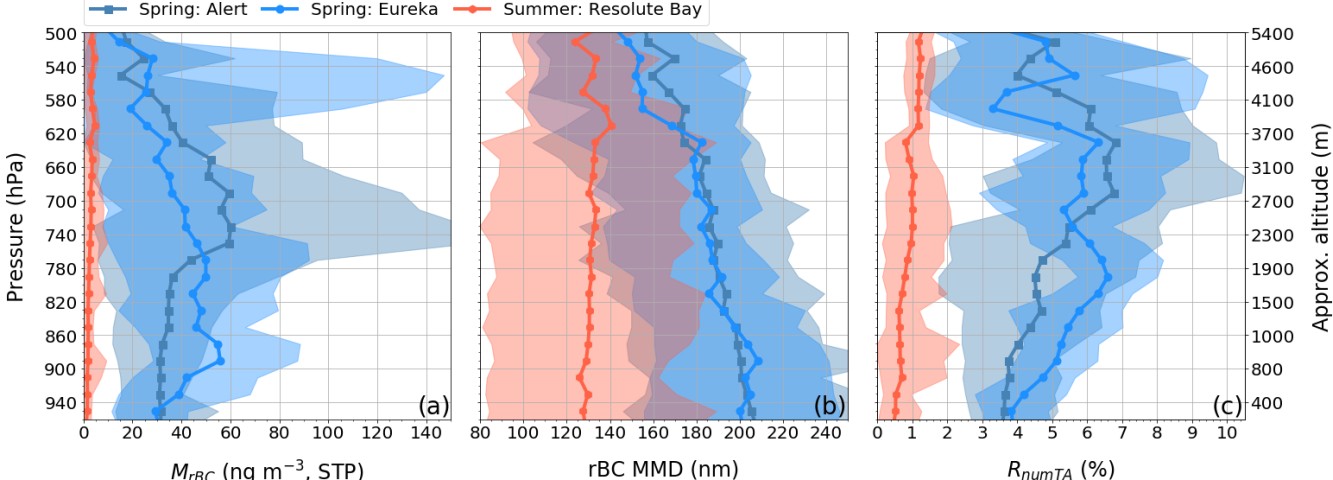

**Figure 4.** Mean regional vertical profiles of (a) rBC mass concentration, (b) mass–mean diameter of rBC particles and (c) ratio of rBC to TA particle number. Shaded areas indicate the minimum 25[th] and maximum 75[th] percentiles of all individual profiles included in the mean profile. The altitude scale may only be used as a guide as it indicates the mean altitude above sea level of each pressure interval for the spring atmosphere.

### 3.2 Seasonal characteristics of rBC vertical distribution in the polar dome

In this section, the vertical distribution of rBC is examined focusing on changes from spring to summer. For each ascent or descent of the flights listed in Tab. 1, data points within fixed pressure-altitude intervals were averaged. These profiles were then successively averaged to mean flight profiles and mean regional profiles, thus avoiding sampling biases due to varying
sampling times in each altitude interval. As shown in Fig. 4, there are substantial differences between the average spring and summer profiles of rBC mass concentration ($M_{rBC}$), mass–mean diameter ($MMD$) and rBC to TA number ratio ($R_{numTA}$).

    The absolute and relative presence of rBC was generally reduced during summer. Ground–based observations at High–Arctic sites like Alert show a pronounced seasonal cycle in rBC concentrations (e.g. Leaitch et al., 2013; Stone et al., 2014; Sharma et al., 2017), which is well matched by the difference in mean $M_{rBC}$ of observations in the lower part of the atmosphere
(>920 hPa) presented here. During spring, averaged $M_{rBC}$ of 31.5 and 30.1 ng m$^{-3}$ were present in the Alert and Eureka region, respectively, while summertime observations showed one order of magnitude lower mean $M_{rBC}$ of 1.4 ng m$^{-3}$. Figure 4a shows that this seasonal difference is present throughout the vertical extent of the polar dome. A one order of magnitude difference in $M_{rBC}$ between the seasons was also found during the ARCTAS spring and summer campaigns in 2008 reported by Matsui et al. (2011). However, their observed $M_{rBC}$ were a factor of two higher compared to the NETCARE observations. This might
be due to the fact that their observations were from a Sub–Arctic region (northern Alaska), where pollution supply and removal not necessarily are in the same balance as within the polar dome. These two observations of pronounced difference in $M_{rBC}$



between the seasons highlights how wet removal becomes more efficient during summer within the polar dome, but as well already during northward transport outside the dome.

During spring, mean profiles from the Alert and Eureka regions showed a similar $M_{\mathrm{rBC}}$ range, however, the $M_{\mathrm{rBC}}$ vertical trend showed certain differences between the two regions. At Eureka, the maxima of the mean profiles occurred between 5   900 and 800 hPa with averaged $M_{\mathrm{rBC}}$ of 55 ng m$^{-3}$ and an interquartile range (IQR) of 11–120 ng m$^{-3}$. Differently, the mean profile from the Alert region reached up to 69 ng m$^{-3}$ at 730 hPa with an IQR between 4 and 157 ng m$^{-3}$ (Fig. 4a). This range of concentrations compares well to the observations by Liu et al. (2015) from northern Scandinavia. These polluted layers, often called Arctic Haze, are however more than a factor 10 lower in $M_{\mathrm{rBC}}$ than observations in pollution plumes close to their sources, e.g. in the mixed source Asian outflow as reported by Kondo et al. (2016), yet are comparable to measurements from 10   the mixed boundary layer over Europe reported by McMeeking et al. (2010).

The profiles of $MMD$ show a nearly steady decrease with altitude from 206 to 162 nm for Alert and 202 to 140 nm for Eureka (Fig. 4b). In contrast, the contribution of rBC particles to the TA by number, $R_{\mathrm{numTA}}$, increased with altitude from around 4% near the ground to 6–7% aloft and $R_{\mathrm{numTA}}$ is particularly variable within altitudes with increased $M_{\mathrm{rBC}}$ (Fig. 4c). Thus, in general, the rBC mass is distributed among higher numbers of smaller sized rBC particles with increasing altitude. 15   Maxima in $R_{\mathrm{numTA}}$ were shifted upwards relative to the nearest maximum in $M_{\mathrm{rBC}}$, which could indicate a partitioning of rBC particle size within polluted layers. High variability in $M_{\mathrm{rBC}}$ occurred above 4000 m in the Eureka region, which can to a large degree be attributed to the variable pressure–altitude of isentropic surfaces that are influenced by pollution transport with variable efficiency – as will be discussed in Sec. 3.3.

In contrast to the spring, the summer $MMD$ show a slight increase with altitude from the surface (129 nm) to about 600 hPa 20   (140 nm; Fig. 4 b). A concentration of rBC mass in small particles could potentially increase the absorption efficiency of the aerosol, but rBC contributed on average only 0.75% by number to the TA (IQR: 0.0–2.1%) and has thus negligible impact on solar light extinction (Fig. 4 c).

## 3.3   Vertical distribution of rBC relative to potential temperature

In order to fully understand the vertical variability of the aerosol distribution, it is important to consider the vertical structure 25   of the polar dome with its core of cold, dense air at the ground and successive dome shaped layers of warmer air above. As shown by Stohl (2006), the transport of air parcels along isentropic surfaces supplies pollutants from lower latitudes to certain levels of the polar dome. The altitude of these surfaces depends on the depth of colder air beneath and varies with the spatial extent of the polar dome induced by synoptic conditions and orographic features. Sampling in different positions relative to the structure of the polar dome has induced variability to the profiles averaged over pressure-altitude intervals. Levels affected by 30   different transport patterns might be unveiled by adopting potential temperature as vertical coordinate, which is monotonically increasing in the stable polar atmosphere.

The measurement periods in both seasons each cover an evolution cycle of a low–pressure system causing a disturbance of the polar dome's structure (see Sec. 3.1), which have caused or altered transport pathways affecting the vertical distribution of rBC. The variability in vertical profiles is evaluated in Sec. 3.3.1 and 3.3.2 for spring and summer, respectively.





### 3.3.1 Vertical distribution of rBC in the spring polar dome

The spring mean flight profiles from Alert and Eureka averaged over intervals of potential temperature are shown in Fig. 5. According to vertical trends and variability patterns in (a) $M_{\mathrm{rBC}}$, (b) $MMD$ and (c) $R_{\mathrm{numTA}}$ as well as (d) the ratio of enhancement over background levels of rBC mass relative to CO mixing ratio ($R_{\mathrm{CO}}$, as defined in Sec. 2.2), five levels can be identified.

The following analysis describes and links patterns in the individual profiles of these properties. The aim is to identify possible implications for mechanisms affecting the aerosol, before source regions and transport patterns for these levels are investigated by the means of kinematic back–trajectories in Sec. 3.4. The vertical boundaries of these levels are confined within strong temperature gradients, which can be found in all profiles, but with slight variation in strength and potential temperature range. However, the vertical location of the gradients varies in relation to pressure–altitude by up to 1000 m between individual pro-

files (Fig. A1).

Figure 5 shows that a homogeneous distribution of rBC with a mean $M_{\mathrm{rBC}}$ of 32 ng m$^{-3}$ (IQR: 13–48 ng m$^{-3}$) and the largest observed $MMD$ of 204 nm (IQR: 153–250 nm) were the characteristics within a temperature gradient capped surface layer with the coldest air between 245 and 255 K (level I). The observed $M_{\mathrm{rBC}}$ across the lowest flight sections in the dome match

well with mean ground–based rBC observations in Alert for spring seasons of the years 2011 to 2013 with $30 \pm 26$ ng m$^{-3}$ reported by Sharma et al. (2017). rBC represented a minor component of the total aerosol by number, with an averaged $R_{\mathrm{numTA}}$ of 3.8%, but together with the large $MMD$ this suggests that the rBC mass is contained in fewer, larger particles. rBC mass was comparably high relative to $\Delta$CO with a mean $R_{\mathrm{CO}}$ of 5.7 ng m$^{-3}$ ppbv$^{-1}$ (IQR: 2.7–10.5 ng m$^{-3}$ ppbv$^{-1}$). This ratio is difficult to compare to observations from literature, because although ranges of values have been attributed to specific

combustion source types, all sampled airmasses were also subject to different ageing time scales leading to significant variations in reported $R_{\mathrm{CO}}$ (Liu et al., 2015). Besides particle removal altering $R_{\mathrm{CO}}$, air pollution from different sources may become mixed in the source region or within the polar dome, thus blending the ratio of rBC to CO. $R_{\mathrm{CO}}$ around 4 to 9 ng m$^{-3}$ ppbv$^{-1}$ were observed by Stohl et al. (2013) at Mt. Zeppelin on Svalbard, European Arctic, in an airmass influenced by transport from northern Russian gas flaring sites. A similar range was observed by Liu et al. (2015) profiling between northern Norway and

Svalbard in spring during transport influence from Europe and Asia. Values up to around 12 ng m$^{-3}$ ppbv$^{-1}$ lay within one standard deviation (Liu et al., 2015). In the middle of this range are observations from Spackman et al. (2008) near fossil fuel combustion sources in Texas with $5.8 \pm 1.0$ ng m$^{-3}$ ppbv$^{-1}$ and observations by Kondo et al. (2016) showed 5.8 ng m$^{-3}$ ppbv$^{-1}$ (IQR: 4.8–7.2 ng m$^{-3}$ ppbv$^{-1}$) in the Asian outflow when flying around 1000 m above sea level. Compared to this, boreal forest fire plumes often show lower ratios (1.2–5.0 ng m$^{-3}$ ppbv$^{-1}$), while grassland and agricultural fires show larger ratios between

10–20 ng m$^{-3}$ ppbv$^{-1}$ (Mikhailov et al., 2017). Due to this large spread of values, it is neither possible to detect measurement environments in which rBC was depleted nor to discriminate source types based on absolute values, however the behaviour of $R_{\mathrm{CO}}$ relative to mean observations within an atmospheric level can give implications for the mechanism changing rBC.

Concentrations and mixing ratios of rBC were increasing throughout a level between about 255 to 265 K (level II). The mean $M_{\mathrm{rBC}}$ was 44 ng m$^{-3}$ (IQR: 15–79 ng m$^{-3}$) and mean $MMD$ was 198 nm (IQR: 143–251 nm). The Eureka profiles (11





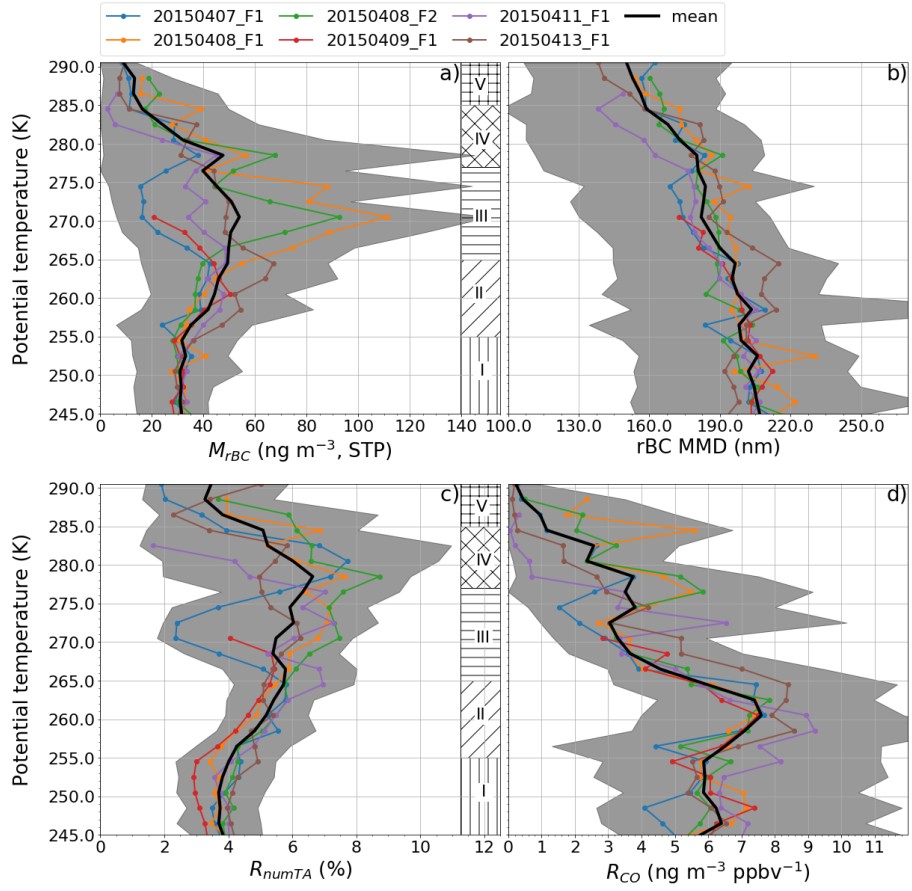

**Figure 5.** Flight profiles averaged over intervals of potential temperature from the spring polar dome of (a) $M_{\text{rBC}}$, (b) $MMD$, (c) $R_{\text{numTA}}$ and (d) $R_{\text{CO}}$. The grey shading around the mean over all profiles (black line) indicates the minimum 25th and maximum 75th percentile of the flight profiles included in the mean. Five levels with different variability patterns as discussed in the text are marked with hatch patterns.

and 13 April) show higher $M_{\text{rBC}}$ at this level than observed around Alert. The highest overall $R_{\text{CO}}$ occurred within this level with a mean of $7.4 \, \text{ng m}^{-3} \, \text{ppbv}^{-1}$ (IQR: $3.8$–$12.0 \, \text{ng m}^{-3} \, \text{ppbv}^{-1}$). Profiles from the Eureka region, especially those of 13 April, showed above average $M_{\text{rBC}}$ together with maxima in $R_{\text{CO}}$ ($8$–$9 \, \text{ng m}^{-3} \, \text{ppbv}^{-1}$) and $MMD$ ($215 \, \text{nm}$), which suggests that either or both, removal along the transport pathway was lower or pollution from a different mix of sources was entrained

5    into the airmass. rBC mass–size distributions (MSD), and therefore also $MMD$, are known to exhibit systematic differences between biomass burning and fossil fuel combustion (Kondo et al., 2011b; Sahu et al., 2012). While the measured $R_{\text{CO}}$ in this level were comparable to the above mentioned literature values from fossil fuel combustion influenced airmasses, the range of observed $MMD$ is more similar to biomass burning plumes. This suggests a complex mixture of rBC from different sources as well as that less efficient wet removal occurred during transport, since wet removal will decrease the $MMD$ (e.g.

10   Kondo et al., 2016). At the same supersaturation, larger BC particles activate nucleation and grow more efficiently into cloud





droplets (Kelvin effect; Seinfeld and Pandis, 2016), which are eventually removed from the atmosphere via precipitation. The supersaturations an air parcel experiences during transport and the formation processes of rain clouds (including collisional processes and the formation of water and ice particles) are spatially inhomogeneous and depend on whether the air parcel gets uplifted or transported at low levels (Jacobson, 2003; Taylor et al., 2014). Ice-cloud scavenging of rBC particles during

transport from mid–latitudes to the Arctic in winter and spring is less efficient than in the liquid phase (Browse et al., 2012). The larger $MMD$ observed on 13 April indicate that there was an absence of nucleation scavenging possibly under clear sky conditions or below cloud level, which made the transport more efficient.

Highest variability in rBC abundance and properties was present between 265 to 277 K (level III). A drop in mean $R_{CO}$ and $MMD$ suggests a different regime in which supply from sources and aerosol removal were in a different balance than

in the levels below. At the beginning of the observation period (7 April), low mean $M_{rBC}$ of 17 ng m$^{-3}$ (IQR: 4–22 ng m$^{-3}$) were measured, while the two flights on 8 April encountered significantly higher concentrations up to 111 ng m$^{-3}$ (IQR: 65–151 ng m$^{-3}$). The overall average concentration was 49 ng m$^{-3}$. The ratios $R_{numTA}$ and $R_{CO}$ as well as $MMD$ were significantly below average on 7 April suggesting that rBC has been depleted by nucleation scavenging, while supply of polluted air set in on 8 April and lasted over the course of the observation period with variable intensity. The Arctic Haze like polluted

layers with highest $M_{rBC}$ were not connected with to high $R_{CO}$. Compared to the lower polar dome, only half as much rBC as CO is transported within the pollution plumes in level III, with a mean $R_{CO}$ of 3.9 ng m$^{-3}$ ppbv$^{-1}$ (1.5–8.0 ng m$^{-3}$ ppbv$^{-1}$).

The potential temperature range 277 to 285 K (level IV) is in the transition zone to the airmass above the dome (Bozem et al., 2018). Within a temperature gradient zone marking the upper boundary of the dome, all profiles peak before sharply decreasing in the airmass above. This transition is also apparent in a gradient of trace gas concentrations (Bozem et al., 2018),

and occurs at slightly varying temperature. The maximum $M_{rBC}$ on 8 April were comparable to that in the level below, but high $R_{CO}$ around 6 ng m$^{-3}$ ppbv$^{-1}$ suggest a different, more efficient transport to this level. $M_{rBC}$ on the higher end of the IQR in level IV (145 ng m$^{-3}$) were encountered by one out of three profile flights on 11 April. The other two profiles included in the mean of that day encountered air depleted in rBC, where low $MMD$ as well as $R_{CO}$ suggest substantial removal of rBC from the airmass by precipitation. However, rBC reaches its maximum contribution to the TA by number of 6.2%.

Bozem et al. (2018) defined the region of potential temperatures higher than about 285 K (level V) to be outside the polar dome due to a strong negative gradient in CO concentrations and stronger connection of transport trajectories to mid–latitudes. At the highest altitudes of the profiling flights, low $M_{rBC}$ (average of 13 ng m$^{-3}$ with IQR: 0–34 ng m$^{-3}$) and a decrease in $MMD$ (average of 155 nm with IQR: 106–191 nm) combined with a low $R_{CO}$ (0.7 ng m$^{-3}$ ppbv$^{-1}$) suggest that polluted air was transported to this level, but scavenged of much of the BC during lifting of the air parcels.

Different transport pathways between the described levels, featuring distinct variability in $M_{rBC}$ and possibly different sources or removal efficiencies, based on the characteristic vertical changes in $R_{CO}$, will be identified in Sec. 3.4.1.



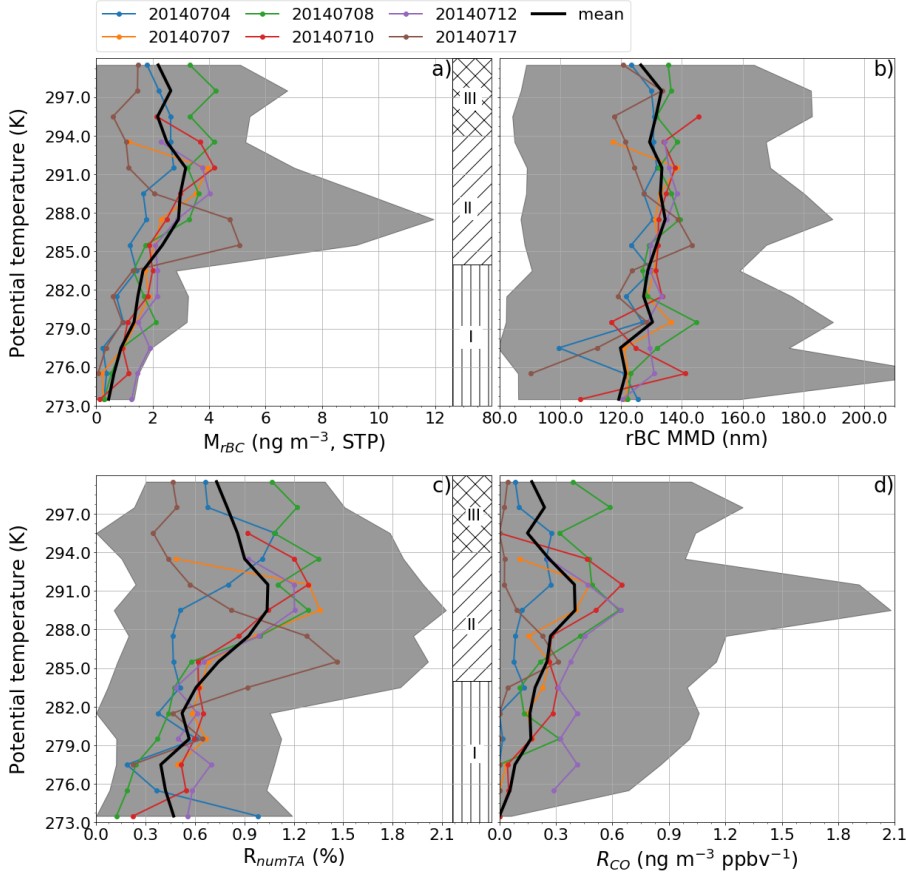

**Figure 6.** Flight profiles averaged over intervals of potential temperature from the summer polar dome of (a) $M_{\text{rBC}}$, (b) $MMD$, (c) $R_{\text{numTA}}$ and (d) $R_{\text{CO}}$. The grey shading around the mean over all profiles (black line) indicates the minimum 25th and maximum 75th percentile of the flight profiles included in the mean. Three levels with different variability patterns, as discussed in the text, are marked with hatch patterns.

### 3.3.2 Vertical distribution of rBC in the summer polar dome

As for the spring case, the variability of aerosol properties as function of the potential temperature was investigated for summer within the polar dome over the area of Resolute Bay (Fig. 6). As already introduced in Sec. 3.2, the general concentration of rBC particles was almost one order of magnitude lower and the variability in the distributions had a lower absolute magnitude than the spring observations. Two strong temperature gradients (Fig. A1) structured the atmosphere below 5 km into three levels in which similar vertical tendencies of rBC concentration and mixing ratios were observed.

Close to the surface, within air at potential temperatures between 273–284 K (level I), the $75^{\text{th}}$ percentile $M_{\text{rBC}}$ did not exceed $3.3\,\text{ng}\,\text{m}^{-3}$. Concentrations in the same order of magnitude ($1\,\text{ng}\,\text{m}^{-3}$) were already observed over the Arctic ocean (Taketani et al., 2016). rBC represented a minor component of the TA throughout the vertical column, however the lowest values were recorded at low altitude, where mean $R_{\text{numTA}}$ was 0.5%. The $R_{\text{CO}}$ well below $1\,\text{ng}\,\text{m}^{-3}\,\text{ppbv}^{-1}$ suggested, combined with the



low particle diameter (average $MMD$ of 125 nm), that particles in the summer polar dome were subject to strong wet removal. The $MMD$ values show a larger variance amongst the different profiles due to few particles in the statistics.

A weakly stable to neutral atmospheric level was present above the stable near–surface level and up to a strong temperature gradient aloft (level II), in which $M_{rBC}$ was relatively constant within the lower part, but increased within the temperature gradient zone in the upper part. This zone lay around 288 to 294 K (Fig. A1) in the period before the weather change (4-12 July) and lower, around 284 to 290 K, on 17 July after the perturbation of the polar dome by a low pressure system (see Sec. 3). The altitude of the gradient zone was changing amongst individual profiles flown in different regions and was likely affected by orography and the variable sea–ice cover (see map in Fig. 1). High humidity was frequently observed in the neutral or weakly stable zone below the temperature gradient. Highest values of $M_{rBC}$ up to $12\,\mathrm{ng\,m^{-3}}$ were encountered around 286 K on 17 July, while in the earlier period, the mean profiles of $M_{rBC}$ peak at only around $4\,\mathrm{ng\,m^{-3}}$, which is however still a factor 2 increase over the concentrations within the less stable lower part of this level. Also the relative presence of rBC showed a significant difference between the two parts. $R_{numTA}$ reached a mean of 1.3% (1.7%) within the concentrations peaks in the first (second) period, while the background in the lower part was around 0.6%. Similarly, $R_{CO}$ was $0.2\,\mathrm{ng\,m^{-3}\,ppbv^{-1}}$ in the background and mean profiles reached $0.6\,\mathrm{ng\,m^{-3}\,ppbv^{-1}}$ within concentrations peaks in the first period. Although the highest rBC concentrations were encountered on 17 July, $R_{CO}$ of 0.0 to $0.3\,\mathrm{ng\,m^{-3}\,ppbv^{-1}}$ indicate that rBC aerosol was depleted compared to the co–emitted CO, which featured elevated concentrations throughout the column compared to the first weather period of stable northern influence (Fig. A1).

The highest investigated level (III) of the atmosphere lay above the upper boundary of the summer polar dome, defined by Bozem et al. (2018) to be situated around 297 to 303 K. Three flights reached this potential temperature level at higher altitudes and show relatively large variability of rBC absolute and relative concentrations, which are within the range of background and elevated concentrations found in the lower levels. Generally, $M_{rBC}$ was higher than at the surface with an IQR of 0.0– $6.7\,\mathrm{ng\,m^{-3}}$.

A back–trajectory analysis will be used below in Sec. 3.4.2 to identify transport patterns and source areas of the summer polar dome for 1) the near–surface layer, 2) the mixed atmosphere and strong temperature gradient zone with increased $M_{rBC}$ as well as 3) the zone above the upper boundary.

## 3.4 Transport patterns and source areas

Kinematic back–trajectories were calculated in order to discern different contributions of potential source regions to the changing characteristics of aerosol properties observed within the potential temperature levels identified above in Sec. 3.3 For their analysis, gridded overpass frequencies were calculated based on the hourly positions of trajectories initiated every 10 seconds along the flight paths, weighted with the $0.5° \cdot 0.5°$ grid area and normalised by the total number of selected back–trajectories for each case. Only trajectories initiated from aircraft positions within the respective levels that encountered above average $M_{rBC}$ were selected and plotted for the analysis in order to increase visibility of pathways with relevant impact on $M_{rBC}$. An exception is level III of the spring polar dome for which the mean of the much lower $M_{rBC}$ encountered on 7 April (see Fig. 5a) was used as selection threshold. Figures 7 and 8 below show the overpass frequencies displayed as heat map overlays. Hatched





are areas where the air moving along the trajectory paths was at high atmospheric pressure and thus likely within the boundary layer, able to pick up pollutants from source regions. Trajectory end points with the location where the air parcels were 10 days prior to release are marked with dots colour coded with $R_{CO}$ measured at the initialisation position.

### 3.4.1 rBC source areas for the spring polar dome

The aerosol over Alert and Eureka in the period 7 to 13 April was influenced by air transport from Eastern Europe and Central Asia as well as North America (Fig. 7). In those regions lobes of cold polar air reached south due to cyclonic perturbations of the polar front (Fig. 2).

Confined by the cyclonic winds around and the calm conditions within the polar dome (Sec. 2), the backward-trajectories initiated in the lowest level (potential temperature 245 to 255 K) showed a long residence time within the Arctic region at low

altitude (Fig. 7 a). Bozem et al. (2018) showed that more than 70% of the trajectories stayed in the polar dome up to 4 days and still 50% did not leave the polar dome 10 days after the initialisation close to Alert. Around 60% of the trajectories initiated around Eureka stayed within the polar dome for 8 days. The limited horizontal extent of this very cold airmass prevented the intrusion of combustion generated aerosol from lower latitudes, leading to relatively low rBC mass concentrations. $R_{CO}$ between 4 to 7 ng m$^{-3}$ ppbv$^{-1}$ were measured when trajectory paths appeared to be re–circulating within the cold air as well

as for paths that had contact with potential sources in the marginal Arctic. This suggests that the dry and stable atmosphere caused long lifetimes of the aerosol, allowing the presence of high numbers of large particles within the aged aerosol ensemble due to inefficient wet removal (Fig. 5 b). Back–trajectories suggest that pollution supply by low-level transport was possible from northern Russia. Correlated with the trajectory overpasses are potential sources such as the nickel mines and smelters in Norilsk (Siberia) and on Kola Peninsula (see annotations in the maps) with their associated marine traffic, which are known

to be high-emitting sources of gases and particulates (Arctic Council, 2009; Stohl et al., 2013; Law et al., 2017; Roiger et al., 2015, and references therein). Also potentially accessible for low–level transport was the oil-rich region of Khanty Mansi southwest of Norilsk where gas flaring emits rBC (Evans et al. (2017); Winiger et al. (2017); see ECLIPSE BC emission inventory in Fig. 7). A low–pressure system facilitated transport from the Great Lakes and the Canadian northeast during parts of the observation period. Pollution might be associated with anthropogenic emissions in populated areas as well as biomass

burning (MODIS active fire data, orange dots in Fig. 7). Pollution within this pattern was not entirely transported at low level, but got lifted before reaching the polar dome at low altitudes.

The observations performed in the two levels between about 255 to 265 K (II) and 265 to 277 K (III) were mainly influenced by airmasses originating over Siberia, Eastern Europe and Central Asia as well as the northwest of the USA and Canada (Fig. 7b and c). Cyclonic perturbations at the polar front induced these transport patterns (Sec. 2) and extended the area influencing

the polar dome airmass, opening low–level transport pathways from wildfires and gas flaring sites (marked in Fig. 7) to the Alert and Eureka regions. The minimal latitude reached by the back–trajectories advanced southward while going to higher potential temperatures from level II to III. However, in order to reach level III above the High–Arctic, air parcels had to be lifted, e.g. in frontal zones or induced by orography. Pathways into the the region from Northern America were lifted within a frontal system but also affected by the Greenland ice sheet, which is reaching up to a mean altitude of about 2 km – an





observation in accordance with general theory (e.g. Stohl, 2006; Quinn et al., 2011). While air parcels from the Eurasian sector travelled at low altitude into level II, level III was not accessible by low–level transport (compare hatched areas in Fig. 7). Despite the higher $M_{rBC}$ observed in level III, there is a strong decrease in $R_{CO}$ from level II to III (Fig. 5). While a shift in $R_{CO}$ can indicate that a different type of combustion source influenced the airmass (as discussed in Sec. 3.3), it is not apparent

from the back–trajectories that a clear shift from one source type to another happened between the two levels. Trajectories pointing to similar origins reached the polar dome in level III with reduced $R_{CO}$, which supports the hypothesis of a regime change towards more efficient wet removal in the upper polar dome (Sec. 3.3).

The origin of air parcels found in level IV between 277 and 285 K showed some differences compared to the lowest atmospheric levels (Fig. 7 d). A number of complex transport patterns occurred at times in association with different parts of the

measurements. For instance, transport possibly from north eastern America, Scandinavian cities and gas flaring sites in the North Sea contributed to elevated $M_{rBC}$ only for observations around Alert. In the Eureka region, re–circulation loops of air that was already within the High–Arctic for a longer time caused peaks in $M_{rBC}$ that were associated with lower $R_{CO}$ values compared to Alert (Fig. 5 a and d). Longer duration of atmospheric processing during the transport might be the cause for decreased $R_{CO}$. The origin of the air parcels is not fully traceable with the 10 days back–trajectories, but supposedly one of the

prominent loops in Fig. 7d originated from wildfires east of Lake Baikal (southern Russia/northern China) with transport facilitated by a low–pressure system (Fig. 2). The air subsided over the Bering and Laptev Seas before being lifted and transported into the polar dome above 3.5 km. Another loop might be connected to the outbreak of polluted air from Eastern Europe that also affected the lower layers. The air was transported over and across the central polar dome and then subsided over northern Canada, from where it was again redirected northward in a cyclone to reach Eureka after being lifted over central Greenland.

Pollution could have also been entrained during exchange with near surface air over northern Canada, although no obvious combustion sources were present within the respective region. A second branch connected to this transport pattern along the polar front from western Canada and Alaska, where among possible pollution sources were wildfires as well as oil and gas extraction related sites like the Alberta tar sands, Prudhoe Bay on the Alaska North Slope and Anchorage (see ECLIPSE emission inventory data marked as yellow dots in Fig. 7).

The air sampled at potential temperatures >285 K in Level V above the polar dome (Bozem et al., 2018) was characterised by the lowest BC values of the spring campaign. It featured clearly different transport patterns compared to the lower atmospheric levels (Fig. 7 e). Generally, the trajectories ending at Alert and Eureka experienced reduced or no contact with the surface north of 60°N, limiting the entrainment from local Arctic sources. The back–trajectories spread out over North America and reached the polar dome after being lifted over the western part of Greenland's ice shield. A fraction of air parcels were in contact with

the surface above the British Islands and the North Sea. All trajectories carried low $M_{rBC}$ with low $R_{CO}$ into level V, indicating that particles were depleted by removal processes during transport.



**Figure 7.** Heat maps of normalised back–trajectory overpass frequencies in each $0.5° \cdot 0.5°$ grid cell (yellow: low; red: high) for the flights of the spring campaign initialised from five potential temperature levels of the polar dome. Hatched are areas in which trajectories were at high atmospheric pressure and exchange with near surface air could have been possible. Dots at the end point of every trajectory 10 days back in time are colour coded with $R_{CO}$ measured in flight near the stations marked with black labels. The map further shows MODIS active fire detections for the period 10 days prior to the first flight until the day of the last flight (orange dots) and known gas flaring sites (yellow dots) from the ECLIPSE emission inventory.





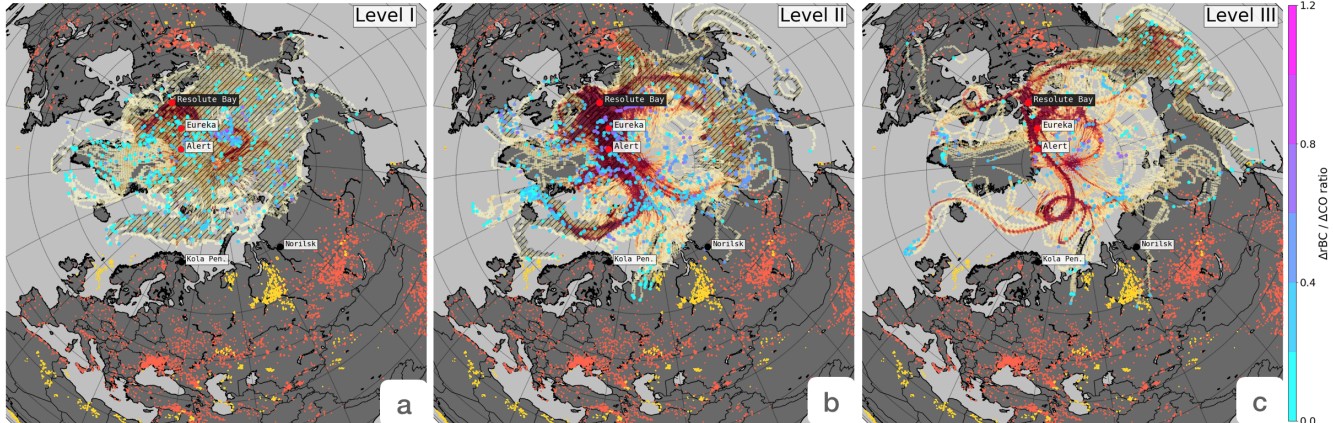

**Figure 8.** Heat maps of normalised back–trajectory overpass frequencies in each $0.5° \cdot 0.5°$ grid cell (yellow: low; red: high) for the flights of the summer campaign initialised from three potential temperature levels of the polar dome. Hatched are areas in which trajectories were at high atmospheric pressure and exchange with near surface air could have been possible. Dots at the end point of every trajectory 10 days back in time are colour coded with $R_{CO}$ measured in flight near the stations marked with black labels. The map further shows MODIS active fire detections for the period 10 days prior to the first flight until the day of the last flight (orange dots) and known gas flaring sites (yellow dots) from the ECLIPSE emission inventory.

### 3.4.2 rBC source areas for the summer polar dome

The aerosol features observed in summer and described in Sec. 6 allowed identifying three different atmospheric levels. By means of back–trajectories it is here investigated how aerosol properties and transport mechanisms for Arctic summer conditions are possibly related. In summer, the polar dome retreats northward and confines air movement to the Arctic Ocean and the northern shores of the bordering continents (Fig 8), with limited entrainment of pollution from lower latitudes. In fact, Bozem et al. (2018) calculated that 80% of trajectories from all levels of the polar dome stayed within the cold Arctic airmass.

Most of the backward-trajectories initiated from level I with strong stable stratification up to potential temperatures of $284\,\mathrm{K}$ stayed in the closer area of Lancaster Sound (Fig. 1) and over the Arctic Ocean for at least 10 days (Fig. 8 a). Pollution within this level was aged and subject to aerosol removal processes, leading to the lowest absolute and relative BC concentrations. More Arctic wide air exchange was possible in level II (potential temperature between 284–294 K; Fig. 8 b). The measurements on 17 July were affected by direct transport from the Alaskan North Slope oil and gas fields around Prudhoe Bay established by a low–pressure system traversing eastward south of Resolute Bay after 13 July (Fig. 3). Wildfires in north-eastern Siberia and north-western Canada supplied combustion generated aerosol to this level, of which higher $M_{\mathrm{rBC}}$ remained only in the more stable part of the level, while strong particle removal during transport is generally evident from low mean rBC particle size and mixing ratios (Sec. 3.3.2).

Level III (potential temperatures $> 284\,\mathrm{K}$) was located above the polar dome and featured several long–range transport pathways that carried air from latitudes as far south as 50°N towards Resolute Bay (Fig. 8 c). A large fraction of the air parcels





made contact with near surface air over Alaska, the Bering Sea and the Far Eastern Federal District of Russia. As a likely consequence of the air being lifted up above the polar dome, rBC was mostly found to be depleted relative to CO, possibly due to precipitation events. Due to the heterogeneity of these events, the variability amongst observations, which are available from three days, is large. $M_{\mathrm{rBC}}$ were higher than in level I but did not exceed concentrations in level II (Fig. 6).

## 3.5 Vertically changing rBC size distribution

As already discussed in Sec. 3.2, a consistent change in the mean diameter of rBC particles was observed between summer and winter (Fig. 4). However, the $MMD$ provides only qualitative information, hence, in order to investigate the variability of rBC particle size in detail, the mass–size distribution of rBC (MSD) was calculated for the different potential temperature levels presented in Sec. 3.3. Figure 9 shows MSD normalised by the total $M_{\mathrm{rBC}}$ for levels I–V from the spring observations around Alert (A) and Eureka (E) as well as for level II from the summer measurements around Resolute Bay. Levels I and III from the summer case were excluded due to insufficient particle numbers.

The mean $MMD$ showed a general trend towards smaller particles with increasing altitude in spring and variability amongst individual profiles indicated sensitivity of the $MMD$ to different atmospheric processing of the aerosol during transport (Sec. 3.3.1). MSD from higher levels (III and above) had a narrower mode with slightly smaller peak diameter. Furthermore, the distribution was clearly shifted towards smaller particles with decreased contribution of rBC cores with diameters larger than about 300 nm, which were only high in number within the lower levels.

The back–trajectory analysis showed that levels I and II in spring were accessible by low–level transport from the marginal Arctic and also areas in Eurasia and North America influenced by cold spills. One of the potential reasons explaining the change of the rBC diameter is the influence of different source types. For instance, Sahu et al. (2012) reported larger particles with $MMD$ of $200 \pm 17$ nm in biomass burning plumes compared to $181 \pm 10$ nm in fossil fuel combustion plumes (values adjusted to $D_{\mathrm{rBC}}$ of particle density $1.8 \, \mathrm{g \, cm^{-3}}$). The observations presented here showed the largest $MMD$ (204 nm with IQR: 153–250 nm) in level I. This level was more accessible to pollution transport from fossil fuel combustion sources (gas flaring) in the marginal Arctic, while higher levels were increasingly influenced by wildfires at lower latitudes (Fig. 7). The potential impact of fossil fuel burning in level I is not supported by the relatively large diameters observed at the surface, suggesting that atmospheric processing controls the size distribution of rBC. The MSD of aged aerosol within the dome could be shifted towards larger particle diameters, compared to the fresh plumes sampled by Sahu et al. (2012) over California, due to coagulation and scavenging without deposition by ice crystals or super-cooled drops (i.e. followed by sublimation or evaporation), which may re–distribute larger particles from higher to lower levels. Moreover, the observed shift in the MSD and the depleted fraction of accumulation mode rBC particles from levels I-II to levels III-IV might suggest two regimes of particle removal, with larger accumulation mode particles being preferentially removed by nucleation scavenging (Jacobson, 2003; Taylor et al., 2014) at higher levels. Vertical lifting at the polar front or up Greenland's ice sheet was involved in trajectory pathways leading into levels III and above (Sec. 3.4.1), which increased the likeliness of cloud formation. The hypothesis of favourable wet removal of larger rBC containing particles in the upper atmosphere (levels III-IV) might be confirmed by the comparable MSD observed in summer at level II (Fig. 9). In fact, the presence of low level clouds in late spring and beginning



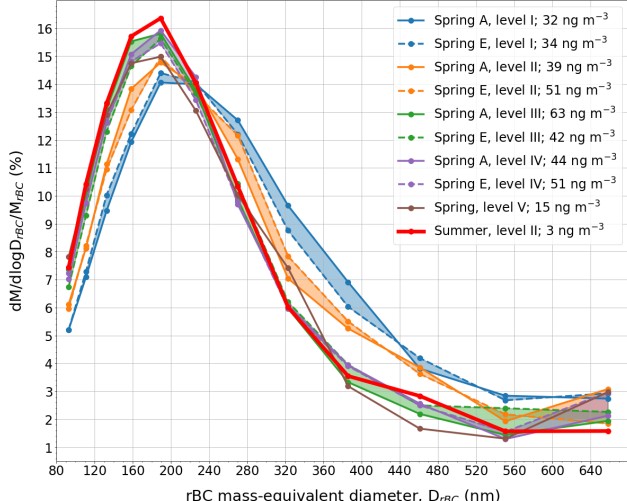

**Figure 9.** Averaged rBC mass size–distributions normalised with the total mass concentration under the curve for levels I–V from the spring polar dome observations around Alert (A) and Eureka (E), as well as for level II from the summer polar dome observations around Resolute Bay.

of summer might result in efficient aerosol wet removal (Garrett et al., 2011; Tunved et al., 2013), which account for 90% of black carbon deposition in the Arctic region (Dou and Xiao, 2016). Moreover, larger rBC particles were observed to be preferentially removed by wet scavenging (Kondo et al., 2016).

Knowing the BC size distribution is a key step in order to asses light absorption by BC. In fact, among other properties, the size of rBC cores affects the mass-absorption cross-section of BC particles (Bond et al., 2006). Within an aged airmass, the particle cores likely get internally mixed with other aerosol. A common assumption is that the BC mass forms the core of a particle and hydrophilic aerosol mass forms a shell around the particle, which then acts as a lens to focus more light onto the core, thereby increasing the absorption efficiency (Kodros et al., 2018, and references therein). Examining the mixing state of rBC particles for the spring campaign in more detail, Kodros et al. (2018) found that the size of rBC containing particles including a coating was on average 1.6 (IQR: 1.4-2.0) times the core diameter for cores around 140 nm and decreased to 1.4 (IQR: 1.2-1.6) times the core diameter at 220 nm. This result is contrasting the observations by Moteki et al. (2012), obtained during the A-FORCE aircraft campaign over the Yellow and East China Seas in spring 2009, that the mass of coating materials is linearly increasing with the mass of rBC cores for aerosol that has not undergone lifting. This suggests that rBC particles with larger hydrophilic coating were preferably activated as cloud condensation nuclei and already scavenged from the aged aerosol observed within the polar dome. Coating thickness was not examined for the summer campaign due to a misalignment of the detector in the SP2 (see Kodros et al., 2018, for technical details), but also due to the generally too low number of rBC particles for the evaluation method. rBC contributed to well below 2% by number to the TA in summer, thus absorption by these particles had negligible contribution to aerosol extinction of solar light, other than in spring, when the contribution varied between 2–10%.



# 4 Summary and conclusions

Two aircraft campaigns within the NETCARE project during spring and summer made it possible to observe the vertically variable distribution of black carbon aerosol over the high–latitude Canadian Arctic.

The seasonality of rBC concentrations found at low altitudes was in good agreement with ground–based observations in Alert (e.g. Sharma et al., 2017) with a one order of magnitude difference in concentrations between spring (mean of $31.6\,\mathrm{ng\,m^{-3}}$) and summer ($1.2\,\mathrm{ng\,m^{-3}}$). This seasonal difference was also apparent in higher atmospheric levels. The vertical variability of rBC was investigated as function of potential temperature, which highlighted that the vertical distribution of aerosols is bound to the vertical structure of the polar dome, the cold and stable airmass over the Arctic. In both seasons, prominent patterns of variability in rBC mass, particle size as well as presence relative to total aerosol number and CO mixing ratio were identified within different vertical levels. The vertical extent of five levels in spring and three levels in summer was connected to the atmospheric stratification. Back–trajectories initialised within each level indicate transport along different isentropic surfaces. The trajectories generally showed that with increasing potential temperature level in the polar dome, air pollution from warmer, more southern areas can affect the High–Arctic. Low–pressure systems at the polar front disturbed the dome and extended the area affecting the levels, which as a consequence increased the number of emission sources entrained into the different levels of the polar dome.

The lowest levels in spring (with potential temperatures $<265\,\mathrm{K}$) were particularly influenced by low–level transport from sources within the marginal Arctic (e.g. gas flaring, mining activities and ships). Between 255 and $265\,\mathrm{K}$, $M_{\mathrm{rBC}}$ gradually increased (to a mean of $49\,\mathrm{ng\,m^{-3}}$) because low–level transport was possible within a cold air outbreak in the corridor from Eastern Europe to Central Asia. Similarly, the temporal development of this disturbance to the polar dome caused an increase of the initially low $M_{\mathrm{rBC}}$ ($<15\,\mathrm{ng\,m^{-3}}$) within the next higher level (between 265 to $276\,\mathrm{K}$). $M_{\mathrm{rBC}}$ peaked at up to $150\,\mathrm{ng\,m^{-3}}$ at altitudes above $2000\,\mathrm{m}$, where rBC and CO from potential gas flaring and biomass burning sources were transported into the dome. Despite the higher rBC concentrations in this level, the rBC to $\Delta$CO ratio decreased to $3$–$4\,\mathrm{ng\,m^{-3}\,ppbv^{-1}}$ from around $7\,\mathrm{ng\,m^{-3}\,ppbv^{-1}}$ in the lower levels. In combination with a shift in the mass–size distributions with mean diameter decreasing from $204\,\mathrm{nm}$ near the ground to $153\,\mathrm{nm}$ at $5\,\mathrm{km}$ above sea level, this further suggests different regimes of particle removal between the lower polar dome, which is low–level transport dominated, and higher levels, where supply by transport likely involves lifting processes and nucleation scavenging. rBC observed above the polar dome around $5$–$6\,\mathrm{km}$ altitude was low in absolute ($<10\,\mathrm{ng\,m^{-3}}$) and relative ($<2\,\mathrm{ng\,m^{-3}\,ppbv^{-1}}$) concentration due to infrequent supply and lifting induced wet removal.

The polar dome of the summer season had little exchange with latitudes south of $70°\mathrm{N}$, which manifested in very low $M_{\mathrm{rBC}}$ ($<3\,\mathrm{ng\,m^{-3}}$) in air colder than $284\,\mathrm{K}$. Observations showed $M_{\mathrm{rBC}} <10\,\mathrm{ng\,m^{-3}}$ as a maximum in the upper part of the summer polar dome, which was affected by transport supplying combustion generated air pollution from wildfires and gas flaring sites located north of $60°\mathrm{N}$. rBC was low relative to $\Delta$CO ($<1\,\mathrm{ng\,m^{-3}\,ppbv^{-1}}$), suggesting that pollution transport to the High–Arctic takes place, but combustion generated aerosol is highly affected by wet removal. The shape of the mean mass–size distribution in summer resembled the spring time mass–size distributions from higher levels, which showed depletion of





particles >300 nm. rBC contributed up to 2–10% to the total aerosol by number in spring. In summer, the contribution decreased well below 2% and thus absorption by rBC particles had negligible effect on aerosol solar light extinction.

The vertical profiles presented here captured the variability in rBC concentrations and properties imposed by cyclonic disturbances to the polar dome over the course of one week in spring and two weeks in summer, which to our knowledge has not been achieved during an aircraft campaign in the High–Arctic before. Moreover, our work represents an extensive dataset of the vertical distribution of rBC in the Arctic, providing new input for the validation of chemical transport models and radiative forcing assessments.

## 5  Data availability

The NETCARE project (Network on Climate and Aerosols: Addressing Key Uncertainties in Remote Canadian Environments, http://www.netcare-project.ca) will make all data publicly available ton the Government of Canada Open Data Portal (https://open.canada.ca/data/en/dataset) in collaboration with Environment and Climate Change Canada. Until then, the data can be accessed by contacting the principal investigator of the network, J. Abbatt, at the University of Toronto (jabbatt@chem.utoronto.ca).

## Appendix A:  Stratification of the polar dome

Figure A1 shows profiles of pressure and CO mixing ratio as function of potential temperature. Strong changes in potential temperature within narrow pressure intervals mark strongly stable stratified zones and hence boundaries between different airmasses in the polar dome. CO mixing ratios had a decreasing trend with altitude/potential temperature in spring while mixing ratios increased in summer. $\Delta$CO is calculated as the difference of measured CO mixing ratios and the altitude dependent background value estimated as the fifth percentile of all measured CO mixing ratios within a potential temperature interval.



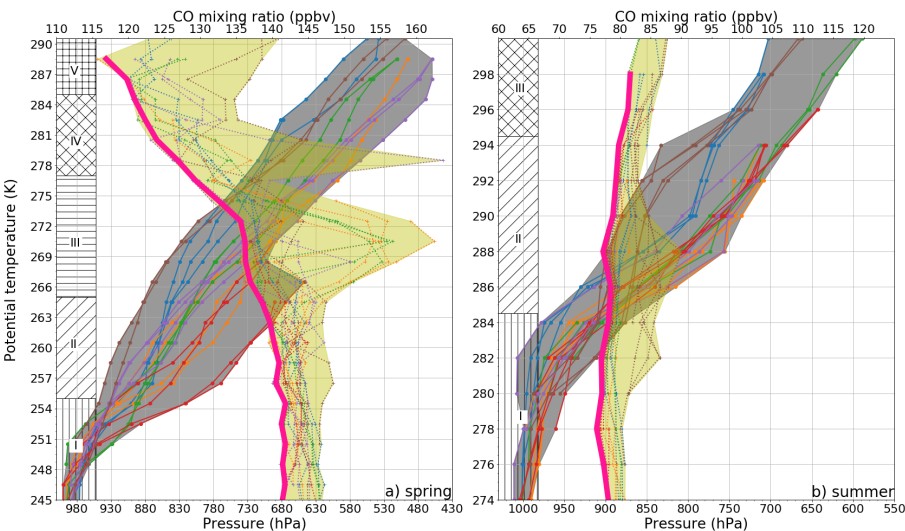

**Figure A1.** Vertical profiles of pressure (lines in grey shading) and CO mixing ratio (lines in yellow shading) versus potential temperature for spring (a) and summer (b). Hatch patterns indicate the extent of the atmospheric levels defined in Sec. A1. The thick pink line marks the fifth percentile of measured CO mixing ratios which is assumed to represent the background concentrations in each potential temperature interval.



*Author contributions.* HS wrote the manuscript, with significant conceptual input from MZ, WRL, ABH, and critical feedback from all co-authors. HS, HB, MDW, JB, and WRL operated instruments in the field and analysed resulting data. HB and PMH ran FLEXPART simulations and HS analysed the resulting data with input from HB. WRL, JPDA and ABH designed the field experiment.

*Competing interests.* The authors declare that they have no conflict of interest.

*Acknowledgements.* This research was funded jointly by NETCARE through the Climate Change and Atmospheric Research (CCAR) program at the Natural Sciences and Engineering Research Council of Canada (NSERC), by the Alfred Wegener Institute (AWI) and by Environment and Climate Change Canada (ECCC). We also gratefully acknowledge the support by the SFB/TR 172 *ArctiC Amplification: Climate Relevant Atmospheric and SurfaCe Processes, and Feedback Mechanisms (AC)*[3] in sub-project 172 funded by the DFG (Deutsche Forschungsgesellschaft). We thank Kenn Borek Air Ltd (KBAL), in particular Gary Murtsell and Neil Travers, as well as Kevin Elke and
John Bayes for their skilful piloting across the Arctic. We acknowledge Martin Gehrmann, Manuel Sellmann and Lukas Kandora (AWI) as well as Doug MacKenzie and Kevin Riehl (KBAL) for their technical help on the ground and during airborne operation. We thank Jim Hodgson and Lake Central Air Services (LCAS) in Muskoka, Jim Watson (Scale Modelbuilders, Inc.), and Julia Binder (AWI) for their support of the instrument integration. We are grateful to Bob Christensen (U. of Toronto), Carrie Taylor, Dan Veber, Alina Chivulescu, Andrew Platt, Ralf Staebler, Anne Marie Macdonald, Desiree Toom and Maurice Watt (ECCC) for their support of the study. We are grateful to
Kathy Law, Jennie Thomas (LATMOS) and Jean-Pierre Blanchet (UQUAM) for their model support during the campaign. Franziska Nehring and Johannes Käsbohrer (FIELAX GmbH) are acknowledged for their support with the meteorological measurements. The meteorological data were supplied by the European Centre for Medium-Range Weather Forecasts (ECWMF). We thank the Nunavut Research Institute and the Nunavut Impact Review Board for licensing the study. Logistical support in Resolute Bay was provided by the Polar Continental Shelf Project (PCSP) of Natural Resources Canada under PCSP Field Project no. 218-14, and we are particularly grateful to Tim McCagherty and
Jodi MacGregor of the PCSP.



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
