# Peer review of "High–Arctic aircraft measurements characterising black carbon vertical variability in spring and summer"

_Atmospheric Chemistry and Physics, 2018_

## Referee Comment (RC1) · Anonymous Referee #1 · 29 Aug 2018

This paper presents vertical distributions of black carbon aerosol from two aircraft campaign in the high arctic during the spring and summer seasons. They look at BC loading, BC fraction of total aerosol, BC mass median diameter and BC/CO and they run back trajectories. The instrumental methods and the writing are fine though the analysis is a rather qualitative and the conclusions basic. The main finding of the paper, as I read it, was that there are seasonal differences in BC sources within and transported to the arctic in the spring and summer that drive marked differences in BC loadings between the two seasons. I don't think that's a particularly surprising finding and the

larger motivations, outlined in the introduction, of connecting these observations to deposition rate to the surface and atmospheric heating are not fully realized. Nor is it a paper that can serve to constrain sources of BC to the arctic; as such I am struggling a bit to define what exactly this paper is about or how it might be used by the community. Below are a few specific instances of where the analysis fell short of conclusive for me.

1. The authors posit (in section 3.2) that the data imply that there is increased wet removal of BC in the summer relative to the spring driving the lower concentrations and smaller size distributions. While this may be true I don't follow the logic of how they have isolated wet deposition from dry deposition within the arctic and different convective processes driving transported airmasses in different seasons. It seems to me that a wide variety of combinations of transport pathways and in-arctic processing could lead to the observed trends.

2. In Section 3.3.1 the authors say "this suggests that the rBC mass is contained in fewer, larger particles." I don't think that this is actually true and I don't know what conclusion we can draw from this statement without any point of comparison. rBC is stated to be a small fraction of total aerosol (3.8%) but relative to what? I believe that anything over 1-2% is actually a large fraction by number when considering most continental locations and many remote ones as well. The MMD observed is rather large but not dramatically larger than observed in other studies for some biomass burning emisisons, including residential burning or for inefficient fossil fuel combustion. I don't think there are a great number of published size distributions for BC aerosol produced from shipping sources, especially in open waters so they could be from that, for example.

3. I'm not sure I see the utility of the labored analysis of the BC/CO relationship. As noted by the authors, sources themselves are known to have high variability in this ratio due to different combustion efficiency and the behavior of this ratio, even for a single source, as a function of processing time, is not well characterized. I think it's probably true that if BC and CO are uncorrelated, one can assume that BC has undergone a long period of transport but trying to make sense of a non-zero number to tell you things

about unknown sources and unknown processing is hard unless you know something about either the source or the processing. In this case the authors seem to be trying to make statements about both at the same time and I'm not sure the data can really tell us much.

4. In all of the discussion of the vertical distributions of BC properties in the springtime polar dome I can't summarize what is learned. Looking at Figure 5, I would guess that the springtime arctic in levels I and II is most impacted by local sources because the MMD is large and the BC/CO ratio is relatively high. Local biomass burning sources and possibly ship emissions could have those characteristics but I don't know if the expected sources are large enough to explain the observed loadings. At the end of this section I guess the conclusion is that different sources are contributing to BC loadings at different atmospheric levels but I don't know what the community can do with that information as presented.

5. In section 3.4, I don't understand why the authors choose to only run back trajectories for times associated with higher concentration observations. It seems that it would be more conclusive if you were to compare back trajectories across a range of observed BC loadings and see which patterns emerge that are common to the high and low concentrations observations respectively.

6. In section 3.4.1 on page 19, there is a mention of the data implying "more efficient wet removal in the upper polar dome". But, again, I think it could also be that the convective processes that lifted the air parcel near the sources were different. What does this really tell us about the polar dome that is useful? Similarly, in section 3.5 the authors try to tie differences between layers to differences in local removal within the arctic but I don't know how local effects can be resolved distinct from source differences and transport effects.

7. On page 23 there is a brief discussion of how MMD might affect the radiative properties of BC in the arctic. Yet the present study doesn't actually report BC coatings or

hydrophilicity so this paragraph serves mostly as a literature review which is an odd note to end the results section. Also, just because BC contributes a small amount of aerosol by number doesn't mean it can't be an important absorber. I agree with the authors that it is probably a negligible contribution to extinction but absorption is what you're really interested in and you have very little information about possible absorption enhancements for these particles.

Smaller comments include:

P9, line 7: "southern boarder" should be "southern border"

P17, line 9: "Highest values" should be "The highest values"

P18, line 31: "The minimal latitude" should be "The minimum latitude"

P18, line 34: "Which is reaching up to" should be "Which reaches"

Towards the end of the paper the authors have several instances of the acronym MSD instead of MMD. Please make the terminology consistent.

P22, line 32: "Likeliness" should be "likelihood"

---

## Referee Comment (RC2) · Anonymous Referee #2 · 30 Aug 2018

This paper presents novel data of black carbon (BC) aerosols obtained by aircraft observations over the Canadian Arctic in summer 2014 and spring 2015. The authors report clear seasonal variations of BC mass concentrations, mass-mean diameter, BC/total aerosols number ratio, and BC/CO ratio, and then try to understand the mechanisms that controlled vertical profiles of BC properties based on the analysis of potential temperature and back-trajectory. The instrumentation is clearly written in the Method section. Because the data of the vertical profiles of BC in the Arctic is very important to assess their radiative impacts and validate numerical models, the novel data presented in this paper is valuable to the community.

[Figure]

One aspect of the paper that I thought could be improved is the discussion on the mechanisms controlling vertical profiles. In many parts of the paper, the authors try to interpret the profile variations by "wet removal" or "nucleation scavenging". However, their conclusion is rather speculative and not well supported by data. I understand that the wet removal process is complex (especially ice-cloud scavenging) and difficult to examine, but if the authors can use any supporting data such as precipitation amount or humidity that air masses had experienced along the back-trajectory paths, more quantitative discussion on the observed variability (and also related new insights) may be achieved in the paper. In addition to this general suggestion, specific comments are given below.

Specific comments:

P6, Line 5: "avalanche photo-diode" should be "photomultiplier tube"?

P7, Line17: "STP" is used here, but defined later in Line 18.

P8, Line 31–33: I could not fully understand what the authors meant here (i.e., the meaning of "threshold filtering"). Please clarify this sentence.

P11, Figure 4: I suggest adding BC/CO data to Figure 4 so that it can correspond to Figure 5.

P12, Line 15: I could not understand what the "partitioning of rBC particle size within polluted layers" meant. Please clarify it.

P14, Figure 5 and 6: In Figure 5, are the colored profiles examples chosen from all the spring fights? (because they are indicated as like "YYYYMMDD_F1" in the legend). On the other hand, in Figure 6, the colored profiles are simply indicated as "YYYYMMDD". Is there any difference?

P15, Line12–13, The low R_numTA does not necessarily suggest that rBC has been depleted by nucleation scavenging, because total aerosols can be also depleted by nucleation scavenging and thus modify the R_numTA value.

P17, Line 18–19: In Figure 6, the level (III) starts at 294 K. Is it correct?

P21, Line 2: "Sec. 6" should be "Sec. 3.3.2"?

P22, Line 19: Sahu et al. (2012) reported mass "median" diameter by applying log-normal fit to the observed size distributions. On the other hand, the present paper reports mass "mean" diameter. For comparison of the data, it should be noted that the definition of "MMD" is different between these studies.

P23, Figure 9: X-axis should be log-scale?

P23, Line 4: "asses" should be "assess".

P23, Line 9–15: Moteki et al. (2012) discussed coating volume (mass), not shell/core diameter ratio. Because a BC particle with core of 220 nm and shell/core diameter ratio of 1.4 actually has larger coating volume than a BC particle with core of 140 nm and shell/core diameter ratio of 1.6, the results explained here are not contrasting but rather consistent with Moteki et al. (2012).

---

## Referee Comment (RC3) · Anonymous Referee #3 · 10 Sep 2018

This paper discusses and analyzes the measurements of the vertical distribution of refractive black carbon (rBC) in the high Arctic during the spring and summer measured with a single particle soot photometer (SP2) during the NETCARE project. The mean and variance of the vertical profiles of total rBC mass, mass-mean diameter, and the ratio of rBC to CO and total aerosol number are discussed, along with the changes in transport patters and sources that lead to the distinct vertical layers observed in the profiles.

The data gathered in this campaign helps to fill an important gap in previous observations of the Arctic. The work presented in the paper is an excellent, detailed analysis

of the sources and mechanisms (e.g., wet deposition) leading to the observed vertical profiles of rBC, helping to provide a conceptual model that explains key features displayed in the observations. Generally, the conclusions of the study are well-justified by the results shown and the potential for alternate explanations is appropriately discussed.

I don't have any major concerns with the methodology, results, or conclusions of the study. Most of my comments below focus on either unclear wording in the manuscript text or issues with the figures that made it difficult to identify some of the features discussed in the text. I've only focused on cases where it was unclear to me what the authors intended to say. There are numerous other language choices that struck me as odd, but rather than list them here I would suggest that these be addressed by an English language copy-editor before publication.

Minor concerns:

P1, L3: You might discuss what you mean by "high Canadian Arctic" here to clarify the region your results apply to.

P1, L6: "caused and changed" is an odd phrase. I think you mean the cyclonic disturbances caused additional transport of pollution, correct?

P3, L5 "spread of more than one order of magnitude" in what? rBC concentrations, deposition, or emissions? Please clarify what you are referring to here.

P5, Table 1: Please add a note to the table explaining that the "station" is the location the plane left from.

P10, Figures 2 and 3: The black triangles are very hard to see against the blue and green colors. Consider making the triangles white instead?

P11, Figure 4: The two blue shades for Alert and Eureka are very hard to tell apart. Can you make them easier to distinguish?

P12, L1-2: I'm not sure the ARCTAS and NETCARE observations discussed in this paragraph are enough to conclude that "wet removal becomes more efficient during summer within the polar done, but as well already during northward transport outside the dome." Do you have other evidence supporting that the changes seen in both campaigns are due to more efficient wet removal?

P12, L7-9: Are you saying that mixed Asian outflow is a source of the Arctic haze you observed in this campaign, or is it just an example to show that the haze concentration is usually lower than near the source?

P12, L15-16: I'm not clear what you mean by "could indicate a partitioning of rBC particle size within polluted layers." Can you please clarify what you mean by this sentence?

P12, L21-22: I think you need to be careful with the writing here. rBC could have a significant impact on solar light extinction (measured in W/m2) even if the number fractions of rBC particles relative to total aerosol was low. However, the low ratio, combined with the low rBC mass concentration, means the impact in this case is negligible. Thus, I think you have to mention the low mass concentration of rBC here before concluding rBC has a negligible impact.

P17, L31-32: How does this choice of only using trajectories that encountered above average M_rBC potentially impact your analysis and results? Is there a potential for bias from this choice?

P18, L1-2: Why did you not use the ECMWF boundary layer heights to determine the hatching instead?

P19, L1-2: I don't see the difference between Levels II and III discussed in this sentence in Figure 7. What should I be looking for in the figure?

P20, Figure 7 and P21, Figure 8: These figures are very difficult to see, maybe due to their small size in the combined figure. The color bar for the overpass frequencies

should also be included in both figures. I'm also wondering if the color for high rBC/CO values is too similar to the red colors used for high overpass frequencies, so in Figure 7 a, I'm not sure if the red near the surface sites is just high overpass frequencies or also high rBC/CO ratios at the endpoints.

Typos and other style issues:

P1, L4: expand the acronym "NETCARE" here.

P1, L10: "factor of 10", not "factor 10"

P1, L22: "rBC was affected" not "got affected"

P7, L32: "As opposed to aerosol", not "Other than aerosol"

P21, L2: There isn't a Section 6, so what should this refer to?

P25, L10: "available ton the Government"?

---

## Referee Comment (RC4) · Anonymous Referee #4 · 12 Sep 2018

General comments;

This paper describes the results from the aircraft measurements of black carbon (BC) aerosols over the high arctic region. The vertical distribution of BC is one of the most important characteristics for assessing its radiative impact. Authors analyzed in detail the vertical distributions, their seasonal variations, and transport pathways of BC using the data sets from the aircraft observations which were performed in the summer of 2014 and the spring of 2015. The analyses of the vertical distribution of BC with potential temperature illustrated the fundamental feature of the transport of BC from the lower latitudinal region (i.e., Sub-Arctic). Single particle soot photometer (SP2) was deployed on the aircraft to reveal one of the microphysical parameters, size distributions, of BC. The changes in the size distributions of BC in the vertical coordinate indicated that the removal process of BC during the transport to the high-arctic region is related to precipitation. The results and discussion presented in this study meet the scope of ACP. The observed features, which are well illustrated in this study, will be really helpful for the research community of Arctic climate changes as well as I actually enjoyed reading this paper.

What this paper does not present in detail is the analyses of wet removal process of BC during the transport and its impact on the abundance and microphysical parameters of BC-containing particles. The cloud processing and following precipitation during the transport in East Asia can significantly affect the microphysical parameters of BC-containing particles in the lower free troposphere (Moteki et al., 2012; Kondo et al., 2016) and even in the planetary boundary layer over the outflow area (Miyakawa et al., 2017). There should be a difference in the actual wet removal process between East Asia and Arctic, because the scavenging of BC particles can be affected by cloud phase (e.g., Browse et al., 2012). Furthermore, we are interested in where BC-containing particles were removed and deposited in Arctic region in order to well understand the snow darkening induced by deposited BC. The more data analyses of precipitation during the transport (intensity of precipitation, where air masses were affected by precipitation, etc.) magnify the significance of the data sets used in this study.

Other minor comments are shown as follows.

Minor comments;

P1, L10. "a factor 10" should be "a factor of 10". P15, L2. "an air parcel" should be "in air parcel". P23, L11-13. The finding in Moteki et al. (2012) is that the average mass of non-BC materials on rBC-containing particles increased with increasing rBC core diameters. They just discussed shell to core (S/C) ratio of rBC-containing particles.

[Figure]

When we translate the relative enhancement of shell mass of non-BC materials into the S/C ratio, the similar tendency given in Kodros et al. (2018) will also be found in Moteki et al. Please modify this description and add appropriate discussion on this part.

References;

Browse, J., Carslaw, K. S., Arnold, S. R., Pringle, K., and Boucher, O.: The scavenging processes controlling the seasonal cycle in Arctic sulphate and black carbon aerosol, Atmos. Chem. Phys., 12, 6775-6798, https://doi.org/10.5194/acp-12-6775-2012, 2012.

Kodros, J. K., Hanna, S. J., Bertram, A. K., Leaitch, W. R., Schulz, H., Herber, A. B., Zanatta, M., Burkart, J., Willis, M. D., Abbatt, J. P. D., and Pierce, J. R.: Size-resolved mixing state of black carbon in the Canadian high Arctic and implications for simulated direct radiative effect, Atmos. Chem. Phys., 18, 11345-11361, https://doi.org/10.5194/acp-18-11345-2018, 2018.

Kondo, Y., N. Moteki, N. Oshima, S. Ohata, M. Koike, Y. Shibano, N. Takegawa, and K. Kita, Effects of wet deposition on the abundance and size distribution of black carbon in East Asia, J. Geophys. Res. Atmos., 121, 4691–4712, doi:10.1002/2015JD024479, 2016.

Miyakawa, T., Oshima, N., Taketani, F., Komazaki, Y., Yoshino, A., Takami, A., Kondo, Y., and Kanaya, Y.: Alteration of the size distributions and mixing states of black carbon through transport in the boundary layer in east Asia, Atmos. Chem. Phys., 17, 5851-5864, https://doi.org/10.5194/acp-17-5851-2017, 2017.

Moteki, N., Kondo, Y., Oshima, N., Takegawa, N., Koike, M., Kita, K., Matsui, H., and Kajino, M.: Size dependence of wet removal of black carbon aerosols during transport from the boundary layer to the free troposphere, Geophys. Res. Lett., 39, L13802, doi:10.1029/2012GL052034, 2012.

---

## Author Comment (AC1) · 7 Dec 2018

–Arctic aircraft measurements characterising black carbon vertical variability in spring and summer

We would like to thank the referees for their detailed and constructive comments, which helped us to improve our manuscript.

For easier reading, we attached our comments as PDF, where the referee comments

are given in black bold, our answers are given below in blue letters. Additionally, we added the changes we made in the revised manuscript in blue bold letters.

Answers of the authors to anonymous Reviewer1

*Anonymous Review of Manuscript acp-2018-587 GENERAL REMARKS:*
*This paper presents vertical distributions of black carbon aerosol from two aircraft campaign in the high arctic during the spring and summer seasons. They look at BC loading, BC fraction of total aerosol, BC mass median diameter and BC/CO and they run back trajectories. The instrumental methods and the writing are fine though the analysis is a rather qualitative and the conclusions basic. The main finding of the paper, as I read it, was that there are seasonal differences in BC sources within and transported to the arctic in the spring and summer that drive marked differences in BC loadings between the two seasons. I don't think that's a particularly surprising finding and the larger motivations, outlined in the introduction, of connecting these observations to deposition rate to the surface and atmospheric heating are not fully realized. Nor is it a paper that can serve to constrain sources of BC to the arctic; as such I am struggling a bit to define what exactly this paper is about or how it might be used by the community.*

The authors would like to point out that the referees raised questions concerning the interpretation of the BC/CO ratio as indicator for wet scavenging and encouraged us to verify the subsequent hypothesis and conclusions. Due to the high number of comments on this specific topic, we prefer to provide here a general and common answer to all reviewers. As a consequence of the above-mentioned reasons, Section 3.4 was substantially modified. The discussion now focusses on the importance of transport patterns on the observed BC concentration. Thus, Figure 7 and Figure 8 were modified. The discussion on potential impact of wet scavenging on BC and

BC/CO ratio is now substantially reduced. However, additional analysis of back trajectories, including encounter with clouds, is now presented in the supplementary material.

Specific comments of Reviewer1

**Please find our comments in the supplementary material to this AC!**

Please also note the supplement to this comment:
https://www.atmos-chem-phys-discuss.net/acp-2018-587/acp-2018-587-AC1-supplement.pdf

[Figure]

**Supplement:**

**High–Arctic aircraft measurements characterising black carbon vertical variability in spring and summer**

We would like to thank the referees for their detailed and constructive comments, which helped us to improve our manuscript. While the referee comments are given in **black bold,** our answers are given below in blue letters. Additionally, we added the changes we made in the revised manuscript in **blue bold** letters.

**Answers of the authors to anonymous Reviewer#1**

*Anonymous Review of Manuscript acp-2018-587 GENERAL REMARKS*

**This paper presents vertical distributions of black carbon aerosol from two aircraft campaign in the high arctic during the spring and summer seasons. They look at BC loading, BC fraction of total aerosol, BC mass median diameter and BC/CO and they run back trajectories. The instrumental methods and the writing are fine though the analysis is a rather qualitative and the conclusions basic. The main finding of the paper, as I read it, was that there are seasonal differences in BC sources within and transported to the arctic in the spring and summer that drive marked differences in BC loadings between the two seasons. I don't think that's a particularly surprising finding and the larger motivations, outlined in the introduction, of connecting these observations to deposition rate to the surface and atmospheric heating are not fully realized. Nor is it a paper that can serve to constrain sources of BC to the arctic; as such I am struggling a bit to define what exactly this paper is about or how it might be used by the community.**

The authors would like to point out that the referees raised questions concerning the interpretation of the BC/CO ratio as indicator for wet scavenging and encouraged us to verify the subsequent hypothesis and conclusions. Due to the high number of comments on this specific topic, we prefer to provide here a general and common answer to all reviewers. As a consequence of the above-mentioned reasons, Section 3.4 was substantially modified. The discussion now focusses on the importance of transport patterns on the observed BC concentration. Thus, Figure 7 and Figure 8 were modified. The discussion on potential impact of wet scavenging on BC and BC/CO ratio is now substantially reduced. However, additional analysis of back trajectories, including encounter with clouds, is now presented in the supplementary material.

*Specific comments of Reviewer#1*

**Below are a few specific instances of where the analysis fell short of conclusive for me.**

Due to its complexity, some comments were split into two or more parts. This allowed responding to the specific issues and making our answers clearer.

**1)The authors posit (in section 3.2) that the data imply that there is increased wet removal of BC in the summer relative to the spring driving the lower concentrations and smaller size distributions. While this may be true I don't follow the logic of how they have isolated wet deposition from dry deposition within the arctic and different convective processes driving transported airmasses in different seasons. It seems to me that a wide variety of combinations of transport pathways and in-arctic processing could lead to the observed trends.**

This issue was indicated by all reviewers. We are now aware of the limits of the interpretation based on our data. The discussion on potential wet removal is now supported by an enlarged set of references, while additional back trajectory data were analysed in order to better understand any influence of wet removal. However, separating wet from dry removal or identifying the main removal mechanism is beyond the scope of the present work. Hence, the discussion on wet removal is now substantially reduced due to the absence of strong evidence supporting the hypothesis.

**2.1) In Section 3.3.1 the authors say "this suggests that the rBC mass is contained in fewer, larger particles." I don't think that this is actually true and I don't know what conclusion we can draw from this statement without any point of comparison.**

We agree with the referee that the statement might be confusing and, in its current form, it appears to indicate that most of the rBC mass was found in larger particles. Although this might be potentially true if there was a pronounced mode of large particles in the size distribution. But, such a situation was not indicated by the analysis of mass—size distributions in Sec. 3.5. The statement was intended to emphasize the smaller number of rBC particles relative to the number of total aerosol compared to upper levels, while the mass-mean diameter was largest in the lowest atmospheric level. As this statement confused the main message, it was removed and the consistent part of the text (P13, L9-16) was reformulated as:

**[…] The profiles in Fig. 5 show a homogeneous distribution of rBC with a mean $M_{rBC}$ of 32 ng m$^{-3}$ (IQR: 13–48 ng m$^{-3}$) that was present in a temperature gradient capped surface layer (level I). This layer held the coldest air encountered with temperatures of 255 K down to 245 K. The observed $M_{rBC}$ across the lowest flight sections matches well with the mean ground–based rBC observations performed in Alert for spring seasons of the years 2011 to 2013 with 30±26 ng m$^{-3}$ (Sharma et al., 2017). Moreover, level I showed the highest average MMD of 204 nm (IQR: 153–250 nm). Such large mean rBC core diameters were already observed at the surface in the European Arctic in spring (Raatikainen et al., 2015; Zanatta et al., 2018) and are distinctly different from freshly emitted rBC (MMD ≈ 100 nm) in urban areas (Laborde et al., 2013). […]**

**2.2) rBC is stated to be a small fraction of total aerosol (3.8%) but relative to what? I believe that anything over 1-2% is actually a large fraction by number when considering most continental locations and many remote ones as well.**

The number fraction of rBC was calculated over the number concentration of total aerosol measured by the Ultra High Sensitivity Aerosol Spectrometer (UHSAS) in the optical diameter range of 85-1000 nm (see P7, L1-10). The referee is actually right, the particle number concentration provided by the UHSAS is a relatively small fraction of the total particles, especially considering that particles below 100 nm constitutes the majority of the aerosol number concentration. As a consequence, the here presented $R_{numTA}$ represent a higher estimation of rBC number fraction. However, our ratios are lower compared to those published in (Kodros et al., 2018; Raatikainen et al., 2015; Sharma et al., 2017), due to their different restriction of the overlapping size range. Nevertheless, $R_{numTA}$ is a useful proxy for assessing the BC relative presence and for isolating any eventual smoke events. The statement now reads:

**[…] Although rBC represented a minor component of the total aerosol in the respective size range by number, with an averaged $R_{numTA}$ of 3.8% that was low with respect to higher levels (II-V), rBC mass was comparably high relative to co–emitted CO with a mean $R_{CO}$ of 5.7 ng m$^{-3}$ ppbv$^{-1}$ (IQR: 2.7–10.5 ng m$^{-3}$ ppbv$^{-1}$ ). […]**

The description of BC number fraction in Section 2.2.1 was improved:

**[…] In the here presented work, the number ratio of rBC over TA particles, $R_{numTA}$, was used to identify atmospheric layers influenced by combustion generated aerosol. It must be noted that, due to the restricted detection range of the UHSAS, the TA number is biased low and $R_{numTA}$ must be considered as an upper estimate of the number fraction of rBC particles. […]**

**2.3) The MMD observed is rather large but not dramatically larger than observed in other studies for some biomass burning emissions, including residential burning or for inefficient fossil fuel combustion. I don't think there are a great number of published size distributions for BC aerosol produced from shipping sources, especially in open waters so they could be from that, for example.**

Although the diameter range reported in the present study might not be extreme, it is at the upper end of BC diameters observed under different conditions by (Laborde et al., 2013): traffic BC=100 nm; biomass burning BC = 130 nm; aged BC = 160 nm; continental BC = 200 nm. The comparison with other observations is now included in the text, see answer to comment 2.1. Considering the ship emissions, we cannot exclude a priori the influence of large BC emitted by ships. Nevertheless, the frozen sea precludes a consistent presence of vessels in the Canadian Arctic during spring. As a consequence, the local influence of ship emission may be considered negligible. Nevertheless, part of the text in Section 3.4.1 was modified in order to better treat this topic:

**[…] In level I, the highest MMD and $R_{CO}$ might suggest entrainment of pollution from the marginal Arctic which underwent no or inefficient wet scavenging. In fact, high $R_{CO}$ values were already associated in the past with low precipitation during transport to the Arctic (Matsui et al., 2011). On the other hand, longer atmospheric processing undergone by rBC sampled in the highest atmospheric levels might favour wet removal of larger rBC particles due to increased hygroscopicity (Moteki et al., 2012), and thus potentially explains the decrease of MMD from the surface to level V. […]**

Section 3.5 was also modified with the same goal, see answer to comment 7.

**3. I'm not sure I see the utility of the labored analysis of the BC/CO relationship. As noted by the authors, sources themselves are known to have high variability in this ratio due to different combustion efficiency and the behavior of this ratio, even for a single source, as a function of processing time, is not well characterized. I think it's probably true that if BC and CO are uncorrelated, one can assume that BC has undergone a long period of transport but trying to make sense of a non-zero number to tell you things about unknown sources and unknown processing is hard unless you know something about either the source or the processing. In this case the authors seem to be trying to make statements about both at the same time and I'm not sure the data can really tell us much.**

The potential role of BC/CO ratio as indicator of wet removal was largely modified in the current version of the manuscript. Several parts of the text were modified accordingly, especially in Section 3.4. Finally, we might argue that the BC/CO ratio alone is not an appropriate tool to investigate wet removal in remote locations, due to competing factors affecting the relative presence of BC and CO such as dry deposition and source type. A part of section 3.4.1 now reads:

**[…] The vertical profiles plotted in Fig. 5 showed a gradual decrease of $R_{CO}$ with altitude, excluding a sharp enhancement in level II. Assuming $R_{CO}$ as a useful indicator of wet removal, we could argue that transport patterns involving the lifting of air might have caused preferential removal of aerosol via wet scavenging. Such an approach was already used in the past, combined with accumulated precipitation along trajectories to investigate the impact of wet scavenging on BC concentration in the Arctic (Matsui et al., 2011). Nevertheless, $R_{CO}$ might be also affected by emission type. The enhancement of $R_{CO}$ in level II might be predominantly caused by entrainment of pollution emitted by different sources. In fact, the low-level air transport from Eastern Russia struck areas influenced by both, biomass burning and gas flaring. Similarly, the second RCO peak observed in level IV might be caused by a change in source types linked to the extended influence of airmasses coming from Northeast Asia. However, $R_{CO}$ did not show any clear correlation with the transport patterns and with the occurrence of liquid and ice clouds along the trajectories (Fig. S2). Due to the complexity of the transport pathways, potentially entraining pollution from different source types, $R_{CO}$ alone proved itself to be insufficient in order to assess the impact of atmospheric processing on BC variability in the Canadian Arctic in spring. The parallel interpretation of $R_{CO}$ and accumulated precipitation along the trajectories might be a better tool to investigate the impact of wet removal on BC presence in the Arctic, as already proposed by (Matsui et al., 2011). However, a complete investigation on the efficiency of BC removal mechanism is beyond the scope of the present work. […]**

**4. In all of the discussion of the vertical distributions of BC properties in the springtime polar dome I can't summarize what is learned. Looking at Figure 5, I would guess that the springtime arctic in levels I and II is most impacted by local sources because the MMD is large and the BC/CO ratio is relatively high. Local biomass burning sources and possibly ship emissions could have those characteristics but I don't know if the expected sources are large enough to explain the observed loadings. At the end of this section I guess the conclusion is that different sources are contributing to BC loadings at different atmospheric levels but I don't know what the community can do with that information as presented.**

Conclusions based solely on the vertical profiles may be premature, we can, nevertheless, learn some interesting things out from Figure 5. Sources at the margins of the Arctic Ocean, such as those related to resource extraction, contribute to BC concentrations in the lower dome – especially in level II. Subsidence and long-term accumulation in the dome may also contribute to the background concentrations. High-Arctic local sources might not represent a strong contributor to BC atmospheric load. This is particularly true when we consider that the highest BC mass concentration and number fraction was found way above the surface due to influence from long-range transport. Moreover, the strong decrease of MMD with altitude implies that atmospheric processing or different source types play an important role on BC microphysics. This is of high interest for radiative forcing

estimations, since the mass absorption cross-section of BC is also a function of particle diameter (Kodros et al., 2018). BC at low levels can especially cause strong surface radiative forcing. Finally, in Section 3.3.1 we aimed to describe all the features observed during the flights and to intrude all the properties, such as MMD and $R_{CO}$, which will be used for further discussion in Section 3.3.1, 3.4 and 3.5. However, we recognize that the section needs some adjustments in order to improve the readability and understanding of our findings. In order to avoid redundancy and merely bibliographical writing, some parts of the text were shortened ($R_{CO}$ introduction: P13L25-32) or removed (BC-CCN behaviour: P13L10-P14L7).

**5. In section 3.4, I don't understand why the authors choose to only run back trajectories for times associated with higher concentration observations. It seems that it would be more conclusive if you were to compare back trajectories across a range of observed BC loadings and see which patterns emerge that are common to the high and low concentrations observations respectively.**

Following the comments of anonymous reviwer#1 and other referees, the analysis of back trajectories is now extended to all the considered measured points. The modified approach allowed a better interpretation of the results presented in Section 3.4.

**6. In section 3.4.1 on page 19, there is a mention of the data implying "more efficient wet removal in the upper polar dome". But, again, I think it could also be that the convective processes that lifted the air parcel near the sources were different. What does this really tell us about the polar dome that is useful? Similarly, in section 3.5 the authors try to tie differences between layers to differences in local removal within the arctic but I don't know how local effects can be resolved distinct from source differences and transport effects.**

All the referees shared similar doubts on the interpretation of the BC/CO ratio results provided by the authors. As a consequence, a substantial number of sections were modified, this applies to Section 3.4.1 and to Section 3.5 as well.

**7. On page 23 there is a brief discussion of how MMD might affect the radiative properties of BC in the arctic. Yet the present study doesn't actually report BC coatings or hydrophilicity so this paragraph serves mostly as a literature review which is an odd note to end the results section. Also, just because BC contributes a small amount of aerosol by number doesn't mean it can't be an important absorber. I agree with the authors that it is probably a negligible contribution to extinction but absorption is what you're really interested in and you have very little information about possible absorption enhancements for these particles.**

The authors understand the points risen by the reviewer, which are partly shared by the other reviewers. The above-mentioned section was heavily reworked, and now focus mainly on the observed differences in the BC size distribution. A reduced discussion on the potential causes and impact are finally presented, mostly as an outlook. Section 3.4 now reads:

**[...] Especially in the Arctic region, where import of BC with an airmass and cloud formation driven removal were found to be a synergistic process (Liu, Fan, Horowitz, & Levy, 2011), it became clear that the BC core size distribution alone is not sufficient to determine the dominant removal process or source type. Even though the present dataset does not allow a complete decoupling of factors controlling the seasonal and altitudinal change of BC diameter, the latter might influence the BC optical properties and subsequent radiative forcing. The mass absorption cross-section of pure BC varies as function of BC diameter (Bond & Bergstrom, 2006), and a shift from ~200 nm to ~250 nm of $D_{rBC}$ causes a decrease of the mass absorption cross section from 6.7 m2 g-1 to 4.9 m2 g-1 (Zanatta et al., 2018). Though the direct BC forcing in the Arctic is dominated by the absolute BC concentration and mainly affected by BC mixing state (Kodros et al., 2018), the change of the BC core diameter is rarely considered in radiative forcing estimations. [...]**

*Minor comments of Reviewer#1*

**P9, line 7: "southern boarder" should be "southern border".**

Changed.

**P17, line 9: "Highest values" should be "The highest values"**

Changed.

**P18, line 31: "The minimal latitude" should be "The minimum latitude"**

Changed.

**P18, line 34: "Which is reaching up to" should be "Which reaches"**

Changed.

**Towards the end of the paper the authors have several instances of the acronym MSD instead of MMD. Please make the terminology consistent.**

In most part of the manuscript the discussion on the vertical variability of the BC size is based on the vertical profiles of the mean mass diameter (MMD) presented in Figures 4-5-6. In Section 3.5 the discussion is mainly based on the mass size distribution (MSD; Figure 9) averaged within the atmospheric layer of main interest. The use of "MSD" as acronym is, therefore, justified.

**P22, line 32: "Likeliness" should be "likelihood"**

Changed.

REFERENCES

Bond, T. C. and Bergstrom, R. W.: Light Absorption by Carbonaceous Particles: An Investigative Review, Aerosol Sci. Technol., 40(1), 27–67, doi:10.1080/02786820500421521, 2006.

Corbin, J. C., Pieber, S. M., Czech, H., Zanatta, M., Jakobi, G., Massabò, D., Orasche, J., El Haddad, I., Mensah, A. A., Stengel, B., Drinovec, L., Mocnik, G., Zimmermann, R., Prévôt, A. S. H. and Gysel, M.: Brown and Black Carbon Emitted by a Marine Engine Operated on Heavy Fuel Oil and Distillate Fuels: Optical Properties, Size Distributions, and Emission Factors, J. Geophys. Res. Atmospheres, 123(11), 6175–6195, doi:10.1029/2017JD027818, 2018.

Kodros, J. K., Hanna, S. J., Bertram, A. K., Leaitch, W. R., Schulz, H., Herber, A. B., Zanatta, M., Burkart, J., Willis, M. D., Abbatt, J. P. D. and Pierce, J. R.: Size-resolved mixing state of black carbon in the Canadian high Arctic and implications for simulated direct radiative effect, Atmospheric Chem. Phys., 18(15), 11345–11361, doi:https://doi.org/10.5194/acp-18-11345-2018, 2018.

Laborde, M., Crippa, M., Tritscher, T., Jurányi, Z., Decarlo, P. F., Temime-Roussel, B., Marchand, N., Eckhardt, S., Stohl, A., Baltensperger, U., Prévôt, A. S. H., Weingartner, E. and Gysel, M.: Black carbon physical properties and mixing state in the European megacity Paris, Atmos Chem Phys, 13(11), 5831–5856, doi:10.5194/acp-13-5831-2013, 2013.

Liu, J., Fan, S., Horowitz, L. W. and Levy, H.: Evaluation of factors controlling long-range transport of black carbon to the Arctic, J. Geophys. Res. Atmospheres, 116(D4), D04307, doi:10.1029/2010JD015145, 2011.

Matsui H., Kondo Y., Moteki N., Takegawa N., Sahu L. K., Zhao Y., Fuelberg H. E., Sessions W. R., Diskin G., Blake D. R., Wisthaler A. and Koike M.: Seasonal variation of the transport of black carbon aerosol from the

Asian continent to the Arctic during the ARCTAS aircraft campaign, J. Geophys. Res. Atmospheres, 116(D5), doi:10.1029/2010JD015067, 2011.

Moteki, N., Kondo, Y., Oshima, N., Takegawa, N., Koike, M., Kita, K., Matsui, H. and Kajino, M.: Size dependence of wet removal of black carbon aerosols during transport from the boundary layer to the free troposphere, Geophys. Res. Lett., 39(13), L13802, doi:10.1029/2012GL052034, 2012.

Raatikainen, T., Brus, D., Hyvärinen, A.-P., Svensson, J., Asmi, E. and Lihavainen, H.: Black carbon concentrations and mixing state in the Finnish Arctic, Atmos Chem Phys, 15(17), 10057–10070, doi:10.5194/acp-15-10057-2015, 2015.

Sahu, L. K., Kondo, Y., Moteki, N., Takegawa, N., Zhao, Y., Cubison, M. J., Jimenez, J. L., Vay, S., Diskin, G. S., Wisthaler, A., Mikoviny, T., Huey, L. G., Weinheimer, A. J. and Knapp, D. J.: Emission characteristics of black carbon in anthropogenic and biomass burning plumes over California during ARCTAS-CARB 2008, J. Geophys. Res. Atmospheres, 117(D16), D16302, doi:10.1029/2011JD017401, 2012.

Sharma, S., Leaitch, W. R., Huang, L., Veber, D., Kolonjari, F., Zhang, W., … Ogren, J. A. (2017). An Evaluation of three methods for measuring black carbon at Alert, Canada. *Atmospheric Chemistry and Physics Discussions*, *17*(24), 1–42. https://doi.org/10.5194/acp-17-15225-2017

Taketani, F., Miyakawa, T., Takashima, H., Komazaki, Y., Kanaya, Y., Taketani, F., Miyakawa, T., Inoue, J., Kanaya, Y., Takashima, H., Pan, X. and Inoue, J.: Ship-borne observations of atmospheric black carbon aerosol particles over the Arctic Ocean, Bering Sea, and North Pacific Ocean during September 2014, J. Geophys. Res. Atmospheres, 2015JD023648, doi:10.1002/2015JD023648, 2016.

Zanatta, M., Laj, P., Gysel, M., Baltensperger, U., Vratolis, S., Eleftheriadis, K., Kondo, Y., Dubuisson, P., Winiarek, V., Kazadzis, S., Tunved, P. and Jacobi, H.-W.: Effects of mixing state on optical and radiative properties of black carbon in the European Arctic, Atmospheric Chem. Phys. Discuss., 1–33, doi:https://doi.org/10.5194/acp-2018-455, 2018.

---

## Author Comment (AC2) · 7 Dec 2018

–Arctic aircraft measurements characterising black carbon vertical variability in spring and summer

We would like to thank the referees for their detailed and constructive comments, which helped us to improve our manuscript.

For easier reading, we attached our comments as PDF, where the referee comments

are given in black bold, our answers are given below in blue letters. Additionally, we added the changes we made in the revised manuscript in blue bold letters.

Answers of the authors to anonymous Reviewer2

*Anonymous Review of Manuscript acp-2018-587 GENERAL REMARKS:*
*This paper presents novel data of black carbon (BC) aerosols obtained by aircraft observations over the Canadian Arctic in summer 2014 and spring 2015. The authors report clear seasonal variations of BC mass concentrations, mass-mean diameter, BC/total aerosols number ratio, and BC/CO ratio, and then try to understand the mechanisms that controlled vertical profiles of BC properties based on the analysis of potential temperature and back-trajectory. The instrumentation is clearly written in the Method section. Because the data of the vertical profiles of BC in the Arctic is very important to assess their radiative impacts and validate numerical models, the novel data presented in this paper is valuable to the community. One aspect of the paper that I thought could be improved is the discussion on the mechanisms controlling vertical profiles. In many parts of the paper, the authors try to interpret the profile variations by "wet removal" or "nucleation scavenging". However, their conclusion is rather speculative and not well supported by data. I understand that the wet removal process is complex (especially ice-cloud scavenging) and difficult to examine, but if the authors can use any supporting data such as precipitation amount or humidity that air masses had experienced along the back-trajectory paths, more quantitative discussion on the observed variability (and also related new insights) may be achieved in the paper.*

The authors would like to point out that the referees raised questions concerning the interpretation of the BC/CO ratio as indicator for wet scavenging and encouraged us to verify the subsequent hypothesis and conclusions. Due to the high number of

comments on this specific topic, we prefer to provide here a general and common answer to all reviewers. As a consequence of the above-mentioned reasons, Section 3.4 was substantially modified. The discussion now focusses on the importance of transport patterns on the observed BC concentration. Thus, Figure 7 and Figure 8 were modified. The discussion on potential impact of wet scavenging on BC and BC/CO ratio is now substantially reduced. However, additional analysis of back trajectories, including encounter with clouds, is now presented in the supplementary material.

Specific comments of Reviewer2

**Please find our comments in the supplementary material to this AC!**

Please also note the supplement to this comment:
https://www.atmos-chem-phys-discuss.net/acp-2018-587/acp-2018-587-AC2-supplement.pdf

**Supplement:**

**High–Arctic aircraft measurements characterising black carbon vertical variability in spring and summer**

We would like to thank the referees for their detailed and constructive comments, which helped us to improve our manuscript. While the referee comments are given in **black bold,** our answers are given below in blue letters. Additionally, we added the changes we made in the revised manuscript in **blue bold** letters.

**Answers of the authors to anonymous Reviewer#2**

*Anonymous Review of Manuscript acp-2018-587 GENERAL REMARKS*

**This paper presents novel data of black carbon (BC) aerosols obtained by aircraft observations over the Canadian Arctic in summer 2014 and spring 2015. The authors report clear seasonal variations of BC mass concentrations, mass-mean diameter, BC/total aerosols number ratio, and BC/CO ratio, and then try to understand the mechanisms that controlled vertical profiles of BC properties based on the analysis of potential temperature and back-trajectory. The instrumentation is clearly written in the Method section. Because the data of the vertical profiles of BC in the Arctic is very important to assess their radiative impacts and validate numerical models, the novel data presented in this paper is valuable to the community. One aspect of the paper that I thought could be improved is the discussion on the mechanisms controlling vertical profiles. In many parts of the paper, the authors try to interpret the profile variations by "wet removal" or "nucleation scavenging". However, their conclusion is rather speculative and not well supported by data. I understand that the wet removal process is complex (especially ice-cloud scavenging) and difficult to examine, but if the authors can use any supporting data such as precipitation amount or humidity that air masses had experienced along the back-trajectory paths, more quantitative discussion on the observed variability (and also related new insights) may be achieved in the paper.**

The authors would like to point out that the referees raised questions concerning the interpretation of the BC/CO ratio as indicator for wet scavenging and encouraged us to verify the subsequent hypothesis and conclusions. Due to the high number of comments on this specific topic, we prefer to provide here a general and common answer to all reviewers. As a consequence of the above-mentioned reasons, Section 3.4 was substantially modified. The discussion now focusses on the importance of transport patterns on the observed BC concentration. Thus, Figure 7 and Figure 8 were modified. The discussion on potential impact of wet scavenging on BC and BC/CO ratio is now substantially reduced. However, additional analysis of back trajectories, including encounter with clouds, is now presented in the supplementary material.

*Specific comments of Reviewer#2*

**P6, Line 5: "avalanche photo-diode" should be "photomultiplier tube"?**

The referee is right, the statement was consequently changed:

**[…] The incandescence light detector, a photomultiplier tube […].**

**P7, Line17: "STP" is used here, but defined later in Line 18.**

Lines 17 and following now read:

**[…] The SP2 and UHSAS shared one bypass line off the main aerosol inlet and sampled with constant 120 ccm (volumetric) and 50 ccm (at standard temperature and pressure, STP), respectively. […] The rBC mass and**

**number concentrations presented in this study refer to standard temperature and pressure of 273.15 K and 1013.25 hPa, respectively, […]**

**P8, Line 31–33: I could not fully understand what the authors meant here (i.e., the meaning of "threshold filtering"). Please clarify this sentence.**

In the present version of the manuscript no BC or BC/CO threshold was applied to the back trajectories. The text included in P8L31-33 and P9L1-3 now reads:

**[…] The time series of trajectories along the track of the aircraft were correlated with in–situ measurement values, in order to relate individual features in the vertical profiles to an ensemble of trajectories (see Sec. 3.4). Due to the potential influence of wild fires and gas flaring on BC presence in the Arctic region, the spatial distribution of gas flaring sites from the ECLIPSE (Evaluating the Climate and Air Quality Impacts of Short–Lived Pollutants) emission inventory (Klimont et al., 2017; Stohl et al., 2015) and active fires from the MODIS level 2 satellite product (Giglio et al., 2003) were also considered for the interpretation of trajectory pathways. […]**

**P11, Figure 4: I suggest adding BC/CO data to Figure 4 so that it can correspond to Figure 5.**

The BC/CO data were mainly used to understand the potential impact of wet removal on BC load and properties. The mutual dependency of the BC/CO ratio on source type and on wet removal, and thus the transport history of BC laden air to different height levels of the polar dome, makes a clear discussion and univocal interpretation of its seasonal variability quite delicate. We preferred to reduce the potential to confuse the reader with the superimposed variability of the BC/CO ratio when shown as function of pressure—altitude in order to rather discuss it along with potential temperature as vertical coordinate in Sec. 3.3. As a consequence, we decided to maintain Figure 4 as it is.

**P12, Line 15: I could not understand what the "partitioning of rBC particle size within polluted layers" meant. Please clarify it.**

With this sentence we meant to underline that the peaks of rBC mass concentration did not directly imply an enhancement of the rBC number fraction. This might involve different removal mechanisms for different aerosol types during transport. On the other side, other processes as different emission sources might play a role. Due to its speculative character the present statement was removed.

**P14, Figure 5 and 6: In Figure 5, are the colored profiles examples chosen from all the spring fights? (because they are indicated as like "YYYYMMDD_F1" in the legend). On the other hand, in Figure 6, the colored profiles are simply indicated as "YYYYMMDD". Is there any difference?**

During the spring campaign 2015, but not for the summer campaign 2014, two different flights took place on the same day (8 April). The notations "F1" and "F2" refer to the first and second flight of the day respectively. In order to avoid confusion, the notation "Fx" was removed in the legend of Figure 5 for all flights excluding the two occurred on the 8 April. The caption of Table 1 now describes explicitly the meaning of notations "F1" and "F2".

**P15, Line12–13, The low R_numTA does not necessarily suggest that rBC has been depleted by nucleation scavenging, because total aerosols can be also depleted by nucleation scavenging and thus modify the R_numTA value.**

The authors agree with the referee's comment. The statement now reads:

**[...] Highest variability in rBC abundance and its properties was present between 265 to 277K (level III). At the beginning of the observation period (7 April), low mean M_rBC of 17 ngm$^{-3}$ (IQR: 4–22 ngm$^{-3}$ ) was measured, while the two flights on 8 April encountered significantly higher concentrations up to 111 ngm$^{-3}$ (IQR: 65–151 ngm$^{-3}$ ). The overall average concentration of BC in level III was 49 ngm$^{-3}$. The enhancement of black carbon mass concentration, together with a decrease in the mean R_CO and MMD potentially suggests different transport or removal regimes compared to the lower atmospheric levels. The ratios R_numTA and R_CO as well as MMD were significantly below average on the 7 April, while supply of polluted air set in on 8 April and lasted over the course of the observation period with variable intensity. [...]**

**P17, Line 18–19: In Figure 6, the level (III) starts at 294 K. Is it correct?**

The statement directly refers to the work of Bozem et al. (2018), who defined the upper limit of the polar dome during the summer NETCARE campaign within the potential temperature range of 299-303.5 K. In order to improve its clarity, the statement was modified as:

**[...] The highest investigated level (III) of the atmosphere was characterized by potential temperature above 294 K, and most probably represented a strong temperature gradient separating the polar dome from free tropospheric conditions. In fact, Bozem et al. (2018) identified the upper boundary of the summer polar dome in the potential temperature range of 299-303.5 K. [...]**

**P21, Line 2: "Sec. 6" should be "Sec. 3.3.2"?**

Changed accordingly.

**P22, Line 19: Sahu et al. (2012) reported mass "median" diameter by applying lognormal fit to the observed size distributions. On the other hand, the present paper reports mass "mean" diameter. For comparison of the data, it should be noted that the definition of "MMD" is different between these studies.**

The authors agree with the comment of the referee. After major changes implemented to Section 3.5, the "Sahu et al. (2012)" reference is now only used to underline the sensitivity of BC diameter to different sources. The specific part of the updated Section 3.5 now reads:

**[...] BC size distribution was found to be extremely sensitive to both emission type and atmospheric processing. For example, while rBC particle diameters increase when switching from fresh urban emissions to biomass burning emissions (Sahu et al., 2012; Laborde et al., 2013), ship emitted rBC can show a bimodal size distribution characterised by a second mode above 600 nm of D$_{rBC}$ (Corbin et al., 2018). [...]**

**P23, Figure 9: X-axis should be log-scale?**

Changed.

**P23, Line 4: "asses" should be "assess".**

Changed.

**P23, Line 9–15: Moteki et al. (2012) discussed coating volume (mass), not shell/core diameter ratio. Because a BC particle with core of 220 nm and shell/core diameter ratio of 1.4 actually has larger coating volume than a BC particle with core of 140 nm and shell/core diameter ratio of 1.6, the results explained here are not contrasting but rather consistent with Moteki et al. (2012).**

The referee is right. The coating mass of the considered BC-containing particles was calculated assuming a fixed density of 1 g cm$^{-3}$ according to Moteki et al. (2012). Small BC cores (140 nm of diameter) having a core/shell ratio of 1.6 showed a coating mass of 4.4 fg. On the contrary, the coating mass of larger BC cores (220 nm of diameter) showing a core/shell ratio of 1.4 was quantified to be 9.7 fg. Thus, the coating-core mass dependency presented by Moteki et al. (2012) is here confirmed. However, after major changes implemented to Section 3.5, the discussion on mixing state of BC was largely reduced due to its mere literature review character. Now the mixing state is briefly introduced as a factor influencing the optical properties and radiative forcing of BC as:

**[…] Especially in the Arctic region, where import of BC with an airmass and cloud formation driven removal were found to be a synergistic process (Liu et al., 2011), it became clear that the BC core size distribution alone is not sufficient to determine the dominant removal process or source type. Even though the present dataset does not allow a complete decoupling of factors controlling the seasonal and altitudinal change of BC diameter, the latter might influence the BC optical properties and subsequent radiative forcing. The mass absorption cross-section of pure BC varies as function of BC diameter (Bond and Bergstrom, 2006), and a shift from ~200 nm to ~250 nm of $D_{rBC}$ causes a decrease of the mass absorption cross section from 6.7 m$^2$ g$^{-1}$ to 4.9 m$^2$ g$^{-1}$ (Zanatta et al., 2018). Though the direct BC forcing in the Arctic is dominated by the absolute BC concentration and mainly affected by BC mixing state (Kodros et al., 2018), the change of the BC core diameter is rarely considered in radiative forcing estimations. […]**

REFERENCES

Bond, T. C. and Bergstrom, R. W.: Light Absorption by Carbonaceous Particles: An Investigative Review, Aerosol Sci. Technol., 40(1), 27–67, doi:10.1080/02786820500421521, 2006.

Bozem, H., Hoor, P., Kunkel, D., Köllner, F., Schneider, J., Herber, A. B., Schulz, H., Leaitch, W. R., Willis, M. D., Burkart, J. and Abbatt, J. P. D.: Characterization of Transport Regimes and the Polar Dome During NETCARE 2014 and 2015, Atmos Chem Phys Discuss, to be submitted, 2018.

Corbin, J. C., Pieber, S. M., Czech, H., Zanatta, M., Jakobi, G., Massabò, D., Orasche, J., El Haddad, I., Mensah, A. A., Stengel, B., Drinovec, L., Mocnik, G., Zimmermann, R., Prévôt, A. S. H. and Gysel, M.: Brown and Black Carbon Emitted by a Marine Engine Operated on Heavy Fuel Oil and Distillate Fuels: Optical Properties, Size Distributions, and Emission Factors, J. Geophys. Res. Atmospheres, 123(11), 6175–6195, doi:10.1029/2017JD027818, 2018.

Giglio, L., Descloitres, J., Justice, C. O. and Kaufman, Y. J.: An Enhanced Contextual Fire Detection Algorithm for MODIS, Remote Sens. Environ., 87(2), 273–282, doi:10.1016/S0034-4257(03)00184-6, 2003.

Klimont, Z., Kupiainen, K., Heyes, C., Purohit, P., Cofala, J., Rafaj, P., Borken-Kleefeld, J. and Schöpp, W.: Global anthropogenic emissions of particulate matter including black carbon, Atmospheric Chem. Phys., 17(14), 8681–8723, doi:https://doi.org/10.5194/acp-17-8681-2017, 2017.

Kodros, J. K., Hanna, S. J., Bertram, A. K., Leaitch, W. R., Schulz, H., Herber, A. B., Zanatta, M., Burkart, J., Willis, M. D., Abbatt, J. P. D. and Pierce, J. R.: Size-resolved mixing state of black carbon in the Canadian high Arctic and implications for simulated direct radiative effect, Atmospheric Chem. Phys., 18(15), 11345–11361, doi:https://doi.org/10.5194/acp-18-11345-2018, 2018.

Laborde, M., Crippa, M., Tritscher, T., Jurányi, Z., Decarlo, P. F., Temime-Roussel, B., Marchand, N., Eckhardt, S., Stohl, A., Baltensperger, U., Prévôt, A. S. H., Weingartner, E. and Gysel, M.: Black carbon physical properties and mixing state in the European megacity Paris, Atmos Chem Phys, 13(11), 5831–5856, doi:10.5194/acp-13-5831-2013, 2013.

Liu, J., Fan, S., Horowitz, L. W. and Levy, H.: Evaluation of factors controlling long-range transport of black carbon to the Arctic, J. Geophys. Res. Atmospheres, 116(D4), D04307, doi:10.1029/2010JD015145, 2011.

Sahu, L. K., Kondo, Y., Moteki, N., Takegawa, N., Zhao, Y., Cubison, M. J., Jimenez, J. L., Vay, S., Diskin, G. S., Wisthaler, A., Mikoviny, T., Huey, L. G., Weinheimer, A. J. and Knapp, D. J.: Emission characteristics of black carbon in anthropogenic and biomass burning plumes over California during ARCTAS-CARB 2008, J. Geophys. Res. Atmospheres, 117(D16), D16302, doi:10.1029/2011JD017401, 2012.

Stohl, A., Aamaas, B., Amann, M., Baker, L. H., Bellouin, N., Berntsen, T. K., Boucher, O., Cherian, R., Collins, W., Daskalakis, N., Dusinska, M., Eckhardt, S., Fuglestvedt, J. S., Harju, M., Heyes, C., Hodnebrog, Ø., Hao, J., Im, U., Kanakidou, M., Klimont, Z., Kupiainen, K., Law, K. S., Lund, M. T., Maas, R., MacIntosh, C. R., Myhre, G., Myriokefalitakis, S., Olivié, D., Quaas, J., Quennehen, B., Raut, J.-C., Rumbold, S. T., Samset, B. H., Schulz, M., Seland, Ø., Shine, K. P., Skeie, R. B., Wang, S., Yttri, K. E. and Zhu, T.: Evaluating the climate and air quality impacts of short-lived pollutants, Atmos Chem Phys, 15(18), 10529–10566, doi:10.5194/acp-15-10529-2015, 2015.

Zanatta, M., Laj, P., Gysel, M., Baltensperger, U., Vratolis, S., Eleftheriadis, K., Kondo, Y., Dubuisson, P., Winiarek, V., Kazadzis, S., Tunved, P. and Jacobi, H.-W.: Effects of mixing state on optical and radiative properties of black carbon in the European Arctic, Atmospheric Chem. Phys. Discuss., 1–33, doi:https://doi.org/10.5194/acp-2018-455, 2018.

---

## Author Comment (AC3) · 7 Dec 2018

–Arctic aircraft measurements characterising black carbon vertical variability in spring and summer

We would like to thank the referees for their detailed and constructive comments, which helped us to improve our manuscript.

For easier reading, we attached our comments as PDF, where the referee comments

are given in black bold, our answers are given below in blue letters. Additionally, we added the changes we made in the revised manuscript in blue bold letters.

Answers of the authors to anonymous Reviewer3

*Anonymous Review of Manuscript acp-2018-587 GENERAL REMARKS:*
*This paper discusses and analyzes the measurements of the vertical distribution of refractive black carbon (rBC) in the high Arctic during the spring and summer measured with a single particle soot photometer (SP2) during the NETCARE project. The mean and variance of the vertical profiles of total rBC mass, mass-mean diameter, and the ratio of rBC to CO and total aerosol number are discussed, along with the changes in transport patters and sources that lead to the distinct vertical layers observed in the profiles. The data gathered in this campaign helps to fill an important gap in previous observations of the Arctic. The work presented in the paper is an excellent, detailed analysis of the sources and mechanisms (e.g., wet deposition) leading to the observed vertical profiles of rBC, helping to provide a conceptual model that explains key features displayed in the observations. Generally, the conclusions of the study are well-justified by the results shown and the potential for alternate explanations is appropriately discussed. I don't have any major concerns with the methodology, results, or conclusions of the study. Most of my comments below focus on either unclear wording in the manuscript text or issues with the figures that made it difficult to identify some of the features discussed in the text. I've only focused on cases where it was unclear to me what the authors intended to say. There are numerous other language choices that struck me as odd, but rather than list them here I would suggest that these be addressed by an English language copy-editor before publication.*

The authors would like to point out that the referees raised questions concerning the interpretation of the BC/CO ratio as indicator for wet scavenging and encouraged

us to verify the subsequent hypothesis and conclusions. Due to the high number of comments on this specific topic, we prefer to provide here a general and common answer to all reviewers. As a consequence of the above-mentioned reasons, Section 3.4 was substantially modified. The discussion now focusses on the importance of transport patterns on the observed BC concentration. Thus, Figure 7 and Figure 8 were modified. The discussion on potential impact of wet scavenging on BC and BC/CO ratio is now substantially reduced. However, additional analysis of back trajectories, including encounter with clouds, is now presented in the supplementary material.

Specific comments of Reviewer3

**Please find our comments in the supplementary material to this AC!**

Please also note the supplement to this comment:
https://www.atmos-chem-phys-discuss.net/acp-2018-587/acp-2018-587-AC3-supplement.pdf

**Supplement:**

**High–Arctic aircraft measurements characterising black carbon vertical variability in spring and summer**

We would like to thank the referees for their detailed and constructive comments, which helped us to improve our manuscript. While the referee comments are given in **black bold,** our answers are given below in blue letters. Additionally, we added the changes we made in the revised manuscript in **blue bold** letters.

**Answers of the authors to anonymous Reviewer#3**

*Anonymous Review of Manuscript acp-2018-587 GENERAL REMARKS*

**This paper discusses and analyzes the measurements of the vertical distribution of refractive black carbon (rBC) in the high Arctic during the spring and summer measured with a single particle soot photometer (SP2) during the NETCARE project. The mean and variance of the vertical profiles of total rBC mass, mass-mean diameter, and the ratio of rBC to CO and total aerosol number are discussed, along with the changes in transport patters and sources that lead to the distinct vertical layers observed in the profiles. The data gathered in this campaign helps to fill an important gap in previous observations of the Arctic. The work presented in the paper is an excellent, detailed analysis of the sources and mechanisms (e.g., wet deposition) leading to the observed vertical profiles of rBC, helping to provide a conceptual model that explains key features displayed in the observations. Generally, the conclusions of the study are well-justified by the results shown and the potential for alternate explanations is appropriately discussed. I don't have any major concerns with the methodology, results, or conclusions of the study. Most of my comments below focus on either unclear wording in the manuscript text or issues with the figures that made it difficult to identify some of the features discussed in the text. I've only focused on cases where it was unclear to me what the authors intended to say. There are numerous other language choices that struck me as odd, but rather than list them here I would suggest that these be addressed by an English language copy-editor before publication.**

The authors would like to point out that the referees raised questions concerning the interpretation of the BC/CO ratio as indicator for wet scavenging and encouraged us to verify the subsequent hypothesis and conclusions. Due to the high number of comments on this specific topic, we prefer to provide here a general and common answer to all reviewers. As a consequence of the above-mentioned reasons, Section 3.4 was substantially modified. The discussion now focusses on the importance of transport patterns on the observed BC concentration. Thus, Figure 7 and Figure 8 were modified. The discussion on potential impact of wet scavenging on BC and BC/CO ratio is now substantially reduced. However, additional analysis of back trajectories, including encounter with clouds, is now presented in the supplementary material.

*Specific comments of Reviewer#3*

**P1, L3: You might discuss what you mean by "high Canadian Arctic" here to clarify the region your results apply to.**

The "Canadian High Arctic" definition was already adopted in most of the publications resulted from the NETCARE projects (Abbatt et al., 2018). Generally, High Arctic can be defined as the area ensemble located at latitudes higher than 70°N (AMAP, 2015). In the present study the Canadian High Arctic includes the northernmost Canadian research stations which mainly experience Arctic conditions all year long, being included in the Polar Dome both during the cold and warm season. P1L3 was modified as:

**[…] high Canadian Arctic (>70°N). […]**

**P1, L6: "caused and changed" is an odd phrase. I think you mean the cyclonic disturbances caused additional transport of pollution, correct?**

The statement was modified, now it reads:

**[…] The observation periods covered evolutions of cyclonic disturbances which favored the transport of air pollution into the High–Arctic, as otherwise the air mass boundary largely impedes entrainment of pollution from lower latitudes. […]**

**P3, L5 "spread of more than one order of magnitude" in what? rBC concentrations, deposition, or emissions? Please clarify what you are referring to here.**

The statement was clarified, now it reads:

**[…] the balance of these effects in the Arctic can only be estimated in models as accurately as vertical distributions of BC is known. However, profiles of BC concentration show a spread of more than one order of magnitude amongst different state-of-the-art models as well as between models and observations (AMAP, 2015). […]**

**P5, Table 1: Please add a note to the table explaining that the "station" is the location the plane left from.**

Added.

**P10, Figures 2 and 3: The black triangles are very hard to see against the blue and green colors. Consider making the triangles white instead?**

Changed

**P11, Figure 4: The two blue shades for Alert and Eureka are very hard to tell apart. Can you make them easier to distinguish?**

The colors were chosen to clearly distinguish the spring and summer measurements. In order to improve the readability of the lines, the color of the Alert profiles was changed to green.

**P12, L1-2: I'm not sure the ARCTAS and NETCARE observations discussed in this paragraph are enough to conclude that "wet removal becomes more efficient during summer within the polar dome, but as well already during northward transport outside the dome." Do you have other evidence supporting that the changes seen in both campaigns are due to more efficient wet removal?**

Our interpretation of wet removal was found weak or inconsistent by other reviewers. In order to improve the interpretation of our observations, the frequency of liquid and ice cloud encounter by the air parcels during transport was calculated for each flight and discussed in Section 3.4. Even with this additional tool, it was difficult to properly estimate the effective impact of wet removal on BC concentration and its properties. Thus, the interpretation of wet scavenging based on the BC/CO ratio was substantially reduced. The mentioned statement was removed. The previous statement about the balance of rBC supply and removal is stressed by:

**[…] This might be due to the fact that their observations were from a Sub-Arctic region (northern Alaska), where pollution supply and removal are not necessarily in the same balance as within the polar dome. The balance between supply and removal of rBC appears to have a pronounced seasonality, based on the ARCTAS and NETCARE observations. […]**

**P12, L7-9: Are you saying that mixed Asian outflow is a source of the Arctic haze you observed in this campaign, or is it just an example to show that the haze concentration is usually lower than near the source?**

The statement was meant to compare the mass concentration of BC found in the Arctic with other locations, inside and outside the Arctic. However, the comparison with Asian outflow conditions was removed because of its low relevance in the present context. The sentence was modified as following:

**[…] The overall range of BC concentrations is similar to previous spring observations reported for the European Arctic (Liu et al., 2015) and comparable to measurements from the mixed boundary layer over Europe (McMeeking et al., 2010). […]**

**P12, L15-16: I'm not clear what you mean by "could indicate a partitioning of rBC particle size within polluted layers." Can you please clarify what you mean by this sentence?**

Anonymous referee #2 already reported this issue. With this sentence we meant to underline that the peaks of rBC mass concentration did not directly imply an enhancement of the rBC number fraction. This might involve different removal mechanisms for different aerosol types during transport. On the other hand, other processes, as different emission sources, might play a role. Due to its speculative character, the present statement was removed.

**P12, L21-22: I think you need to be careful with the writing here. rBC could have a significant impact on solar light extinction (measured in W/m$^2$) even if the number fractions of rBC particles relative to total aerosol was low. However, the low ratio, combined with the low rBC mass concentration, means the impact in this case is negligible. Thus, I think you have to mention the low mass concentration of rBC here before concluding rBC has a negligible impact.**

We agree with the referee's comment. The text was changed in accordance to other modifications made in Section 3.5.

**[…] In contrast to the spring, the summer MMD showed a slight increase from the surface (129 nm) to about 600 hPa (140 nm; Fig. 4 b). As pointed out in Bond and Bergstrom (2006), the mass absorption cross-section of BC particles depends, also, on the particle's diameter. As a consequence, the concentration of rBC mass in small particles could potentially contribute to the enhancement of the absorption coefficient of the total aerosol. Nevertheless, the low values of (average of 2 ng m$^{-3}$ with IQR 0—12 ng m$^{-3}$ throughout the column) and $R_{numTA}$ (average of 0.75%) makes BC a minor contributor to the total aerosol light extinction. A more detailed description of seasonal and vertical variability of the BC core diameter will be provided in Section 3.5[…]**

**P17, L31-32: How does this choice of only using trajectories that encountered above average M_rBC potentially impact your analysis and results? Is there a potential for bias from this choice?**

Our choice of using a $M_{rBC}$ threshold was also questioned by other referees. Initially we wanted to focus on the most intense plumes, which most likely cause high but local forcing. We now understand the limits of our choice, which might systematically remove air parcels that originated in pristine regions or that experienced precipitation. For this reason, the trajectories presented and discussed in Section 3.4 include all points at which measurements were made.

**P18, L1-2: Why did you not use the ECMWF boundary layer heights to determine the hatching instead?**

The ECMWF data were used to run the LAGRANTO back-trajectories. A flag indicating whether the trajectory was traveling within the boundary layer was however not part of the available model output. Linking the trajectory position at each time step again with the boundary layer height information from datasets (e.g. ERA-interim) would be possible in post—processing, however only as a complicated approach relaying on several assumptions. A detailed evaluation of the trajectories' interactions with the boundary layer was beyond the scope of the maps in Fig. 7 and 8 and we believe that detailed information would have been blurred in these multiple day average maps. The text was changed and now stresses that the hatching highlights the presence of trajectories moving at atmospheric pressures >920 hPa in a grid cell in contrast to trajectories already lifted up from the lower atmosphere.

**[…] A hatching highlights grids where trajectories travelled at atmospheric pressures >920 hPa, which is equal to less than about 0.5 km. Climatological boundary layer heights over Europe are typically <1 km during daytime [Seidel et al., 2012], thus pollution uptake from surface sources may be possible in a well-mixed atmosphere in the hatched areas in contrast to trajectories moving in the upper atmosphere or being lifted already due to vertical motion in synoptic scale systems (Sec. 3.1). […]**

**P19, L1-2: I don't see the difference between Levels II and III discussed in this sentence in Figure 7. What should I be looking for in the figure?**

The authors agree with the referee's comment. The interpretation of the Figure 7b and Figure 7c was not accurate. In fact, back trajectories suggested that the motion of airmasses from the Eurasian sector to Level II and Level III was quite similar. Insights on transport and origin of air parcels higher in $M_{rBC}$ are now more evident after extending the trajectory study to the entire dataset and focusing the discussion on $M_{rBC}$ instead of $R_{CO}$. The entire Section 3.4 was substantially modified, we thus encourage the reviewer to consider the changes implemented in the entire section.

**P20, Figure 7 and P21, Figure 8: These figures are very difficult to see, maybe due to their small size in the combined figure. The color bar for the overpass frequencies should also be included in both figures. I'm also wondering if the color for high rBC/CO values is too similar to the red colors used for high overpass frequencies, so in Figure 7 a, I'm not sure if the red near the surface sites is just high overpass frequencies or also high rBC/CO ratios at the endpoints.**

The authors are aware of the interpretational limits of both Figure 7 and Figure 8. To improve its readability and clarity, some modifications were implemented:

- The location of the reference stations is now symbolized by a large black cross.
- $R_{CO}$ in Figure 7 and 8 was substituted with $M_{rBC}$.
- Two color scales describing the overpass frequency and $M_{rBC}$ were added.
- A legend now indicates the two source types (wild fires and gas flaring) and the airmasses moving at low level.

*Minor comments of Reviewer#3*

**P1, L4: expand the acronym "NETCARE" here.**

Changed.

**P1, L10: "factor of 10", not "factor 10"**

Changed.

**P1, L22: "rBC was affected" not "got affected"**

Changed.

**P7, L32: "As opposed to aerosol", not "Other than aerosol"**

Changed.

**P21, L2: There isn't a Section 6, so what should this refer to?**

Corrected.

**P25, L10: "available ton the Government"?**

Corrected.

**REFERENCES**

Abbatt, J. P. D., Leaitch, W. R., Aliabadi, A. A., Bertram, A. K., Blanchet, J.-P., Boivin-Rioux, A., Bozem, H., Burkart, J., Chang, R. Y. W., Charette, J., Chaubey, J. P., Christensen, R. J., Cirisan, A., Collins, D. B., Croft, B., Dionne, J., Evans, G. J., Fletcher, C. G., Ghahremaninezhad, R., Girard, E., Gong, W., Gosselin, M., Gourdal, M., Hanna, S. J., Hayashida, H., Herber, A. B., Hesaraki, S., Hoor, P., Huang, L., Hussherr, R., Irish, V. E., Keita, S. A., Kodros, J. K., Köllner, F., Kolonjari, F., Kunkel, D., Ladino, L. A., Law, K., Levasseur, M., Libois, Q., Liggio, J., Lizotte, M., Macdonald, K. M., Mahmood, R., Martin, R. V., Mason, R. H., Miller, L. A., Moravek, A., Mortenson, E., Mungall, E. L., Murphy, J. G., Namazi, M., Norman, A.-L., O'Neill, N. T., Pierce, J. R., Russell, L. M., Schneider, J., Schulz, H., Sharma, S., Si, M., Staebler, R. M., Steiner, N. S., Galí, M., Thomas, J. L., Salzen, K. von, Wentzell, J. J. B., Willis, M. D., Wentworth, G. R., Xu, J.-W. and Yakobi-Hancock, J. D.: New insights into aerosol and climate in the Arctic, Atmospheric Chem. Phys. Discuss., 1–60, doi:https://doi.org/10.5194/acp-2018-995, 2018.

AMAP: AMAP Assessment 2015: Black carbon and ozone as Arctic climate forcers. Arctic Monitoring and Assessment Programme (AMAP), Oslo, Norway. vii + 116 pp., [online] Available from: http://www.amap.no/documents/doc/amap-assessment-2015-black-carbon-and-ozone-as-arctic-climate-forcers/1299 (Accessed 7 January 2016), 2015.

Bond, T. C. and Bergstrom, R. W.: Light Absorption by Carbonaceous Particles: An Investigative Review, Aerosol Sci. Technol., 40(1), 27–67, doi:10.1080/02786820500421521, 2006.

Liu, D., Quennehen, B., Darbyshire, E., Allan, J. D., Williams, P. I., Taylor, J. W., Bauguitte, S. J.-B., Flynn, M. J., Lowe, D., Gallagher, M. W., Bower, K. N., Choularton, T. W. and Coe, H.: The importance of Asia as a source of black carbon to the European Arctic during springtime 2013, Atmospheric Chem. Phys., 15(20), 11537–11555, doi:https://doi.org/10.5194/acp-15-11537-2015, 2015.

McMeeking, G. R., Hamburger, T., Liu, D., Flynn, M., Morgan, W. T., Northway, M., Highwood, E. J., Krejci, R., Allan, J. D., Minikin, A. and Coe, H.: Black carbon measurements in the boundary layer over western and northern Europe, Atmos Chem Phys, 10(19), 9393–9414, doi:10.5194/acp-10-9393-2010, 2010.

---

## Author Comment (AC4) · 7 Dec 2018

–Arctic aircraft measurements characterising black carbon vertical variability in spring and summer

We would like to thank the referees for their detailed and constructive comments, which helped us to improve our manuscript.

For easier reading, we attached our comments as PDF, where the referee comments

are given in black bold, our answers are given below in blue letters. Additionally, we added the changes we made in the revised manuscript in blue bold letters.

Answers of the authors to anonymous Reviewer4

*Anonymous Review of Manuscript acp-2018-587 GENERAL REMARKS:*
*This paper describes the results from the aircraft measurements of black carbon (BC) aerosols over the high arctic region. The vertical distribution of BC is one of the most important characteristics for assessing its radiative impact. Authors analyzed in detail the vertical distributions, their seasonal variations, and transport pathways of BC using the data sets from the aircraft observations which were performed in the summer of 2014 and the spring of 2015. The analyses of the vertical distribution of BC with potential temperature illustrated the fundamental feature of the transport of BC from the lower latitudinal region (i.e., Sub-Arctic). Single particle soot photometer (SP2) was deployed on the aircraft to reveal one of the microphysical parameters, size distributions, of BC. The changes in the size distributions of BC in the vertical coordinate indicated that the removal process of BC during the transport to the high-arctic region is related to precipitation. The results and discussion presented in this study meet the scope of ACP. The observed features, which are well illustrated in this study, will be really helpful for the research community of Arctic climate changes as well as I actually enjoyed reading this paper. What this paper does not present in detail is the analyses of wet removal process of BC during the transport and its impact on the abundance and microphysical parameters of BC-containing particles. The cloud processing and following precipitation during the transport in East Asia can significantly affect the microphysical parameters of BC-containing particles in the lower free troposphere (Moteki et al., 2012; Kondo et al., 2016) and even in the planetary boundary layer over the outflow area (Miyakawa et al., 2017). There should be a difference in the actual wet removal process between*

[Figure]

*East Asia and Arctic, because the scavenging of BC particles can be affected by cloud phase (e.g., Browse et al., 2012). Furthermore, we are interested in where BC-containing particles were removed and deposited in Arctic region in order to well understand the snow darkening induced by deposited BC. The more data analyses of precipitation during the transport (intensity of precipitation, where air masses were affected by precipitation, etc.) magnify the significance of the data sets used in this study.*

The authors would like to point out that the referees raised questions concerning the interpretation of the BC/CO ratio as indicator for wet scavenging and encouraged us to verify the subsequent hypothesis and conclusions. Due to the high number of comments on this specific topic, we prefer to provide here a general and common answer to all reviewers. As a consequence of the above-mentioned reasons, Section 3.4 was substantially modified. The discussion now focusses on the importance of transport patterns on the observed BC concentration. Thus, Figure 7 and Figure 8 were modified. The discussion on potential impact of wet scavenging on BC and BC/CO ratio is now substantially reduced. However, additional analysis of back trajectories, including encounter with clouds, is now presented in the supplementary material.

Specific comments of Reviewer4

**Please find our comments in the supplementary material to this AC!**

Please also note the supplement to this comment:
https://www.atmos-chem-phys-discuss.net/acp-2018-587/acp-2018-587-AC4-supplement.pdf

**Supplement:**

**High–Arctic aircraft measurements characterising black carbon vertical variability in spring and summer**

We would like to thank the referees for their detailed and constructive comments, which helped us to improve our manuscript. While the referee comments are given in **black bold,** our answers are given below in blue letters. Additionally, we added the changes we made in the revised manuscript in **blue bold** letters.

**Answers of the authors to anonymous Reviewer#4**

*Anonymous Review of Manuscript acp-2018-587 GENERAL REMARKS*

**This paper describes the results from the aircraft measurements of black carbon (BC) aerosols over the high arctic region. The vertical distribution of BC is one of the most important characteristics for assessing its radiative impact. Authors analyzed in detail the vertical distributions, their seasonal variations, and transport pathways of BC using the data sets from the aircraft observations which were performed in the summer of 2014 and the spring of 2015. The analyses of the vertical distribution of BC with potential temperature illustrated the fundamental feature of the transport of BC from the lower latitudinal region (i.e., Sub-Arctic). Single particle soot photometer (SP2) was deployed on the aircraft to reveal one of the microphysical parameters, size distributions, of BC. The changes in the size distributions of BC in the vertical coordinate indicated that the removal process of BC during the transport to the high-arctic region is related to precipitation. The results and discussion presented in this study meet the scope of ACP. The observed features, which are well illustrated in this study, will be really helpful for the research community of Arctic climate changes as well as I actually enjoyed reading this paper. What this paper does not present in detail is the analyses of wet removal process of BC during the transport and its impact on the abundance and microphysical parameters of BC-containing particles. The cloud processing and following precipitation during the transport in East Asia can significantly affect the microphysical parameters of BC-containing particles in the lower free troposphere (Moteki et al., 2012; Kondo et al., 2016) and even in the planetary boundary layer over the outflow area (Miyakawa et al., 2017). There should be a difference in the actual wet removal process between East Asia and Arctic, because the scavenging of BC particles can be affected by cloud phase (e.g., Browse et al., 2012). Furthermore, we are interested in where BC-containing particles were removed and deposited in Arctic region in order to well understand the snow darkening induced by deposited BC. The more data analyses of precipitation during the transport (intensity of precipitation, where air masses were affected by precipitation, etc.) magnify the significance of the data sets used in this study.**

The authors would like to point out that the referees raised questions concerning the interpretation of the BC/CO ratio as indicator for wet scavenging and encouraged us to verify the subsequent hypothesis and conclusions. Due to the high number of comments on this specific topic, we prefer to provide here a general and common answer to all reviewers. As a consequence of the above-mentioned reasons, Section 3.4 was substantially modified. The discussion now focusses on the importance of transport patterns on the observed BC concentration. Thus, Figure 7 and Figure 8 were modified. The discussion on potential impact of wet scavenging on BC and BC/CO ratio is now substantially reduced. However, additional analysis of back trajectories, including encounter with clouds, is now presented in the supplementary material.

*Specific comments of Reviewer#4*

**P1, L10. "a factor 10" should be "a factor of 10". P15, L2. "an air parcel" should be "in air parcel".**

Corrected

**P23, L11-13. The finding in Moteki et al. (2012) is that the average mass of non-BC materials on rBC-containing particles increased with increasing rBC core diameters. They just discussed shell to core (S/C) ratio of rBC-containing particles. When we translate the relative enhancement of shell mass of non-BC materials into the S/C ratio, the similar tendency given in Kodros et al. (2018) will also be found in Moteki et al. Please modify this description and add appropriate discussion on this part.**

The same issue was highlighted by anonymous referee#2. The text was modified in order to translate our core to shell diameter ratio into mass ratio. As matter of fact, our results are coherent with the findings of Moteki et al. (2012). The mass of coatings was calculated assuming a fixed density of 1 g cm$^{-3}$ (Moteki et al., 2012) and quantified to be 4.4 fg and 9.7 fg for BC cores having diameters of 140 and 220 nm respectively. However, Section 3.5 was significantly modified and, based on other referees' comments, the statement mentioned by the anonymous reviwer#4 was removed.

REFERENCES

Moteki, N., Kondo, Y., Oshima, N., Takegawa, N., Koike, M., Kita, K., Matsui, H. and Kajino, M.: Size dependence of wet removal of black carbon aerosols during transport from the boundary layer to the free troposphere, Geophys. Res. Lett., 39(13), L13802, doi:10.1029/2012GL052034, 2012.